# Polarization structured light 3D depth image sensor for scenes with reflective surfaces

Xuanlun Huang [1,2,4], Chenyang Wu [1,2,4], Xiaolan Xu[2], Baishun Wang[2], Sui Zhang[2], Chihchiang Shen [2], Chiennan Yu[2], Jiaxing Wang[2], Nan Chi [1,3], Shaohua Yu [1,3] & Connie J. Chang-Hasnain [1,2] ✉

Highly reflective surfaces are notorious in the field of depth sensing and three-dimensional (3D) imaging because they can cause severe errors in perception of the depth. Despite recent progress in addressing this challenge, there are still no robust and error-free solutions. Here, we devise a polarization structured light 3D sensor for solving these problems, in which high-contrast-grating (HCG) vertical-cavity surface-emitting lasers (VCSELs) are used to exploit the polarization property. We demonstrate accurate depth measurements of the reflective surfaces and objects behind them in various imaging situations. In addition, the absolute error and effective measurement range are measured to prove the applicability for a wide range of 3D applications. Our work innovatively combines polarization and depth information, opening the way for fully understanding and applying polarization properties in the 3D domain.

Highly reflective surfaces, such as glass, mirror, and water surface, are common scenes in 3D imaging. They often cause irrevocable errors in depth sensing and 3D imaging, such as mirages leading to wrong depth measurements or specular reflection blinding 3D sensors. However, the correct perception of the surrounding environment is critical for many applications, such as robotics, scene reconstruction, and virtual reality. Hence, it is extremely important to manage the problem of reflective surfaces. In recent decades, there are plenty of techniques managing to detect the reflective surfaces, including polarization imaging, active projectors and its fusion with other sensors, and deep learning. However, due to the effects of reflective or multipath noise, most of these techniques will result in erroneous measurements of the reflective surfaces.

Polarization imaging is able to obtain the 3D shape of an object by analyzing several images of reflected intensity at different polarized angles. Its first use to determine the 3D orientations of reflective surfaces dated back to the 1990s[1]. Since then, polarization imaging has been used to reconstruct the shape of transparent[2–4] and specular[5–8] objects. Furthermore, the integration with cues from various depth sensors[9–14] enables polarization imaging to obtain better shape recoveries of the targets. However, polarization imaging needs to capture multiple polarized images and can only estimate relative depths. In addition, most of them requires knowledge of the refractive index of the object, while some rely on other factors to ease this need, such as various light conditions[15,16] or multiwavelength[17]. In[11], Berger combined polarization and stereo vision and further extended the application to general environments with reflective surfaces. However, due to the inability to differentiate mirages from real objects, stereo vision will result in a virtual and wrong distance. Some methods use multiple polarized images[18–22] or local reflection cues[23,24] to separate the reflection. In some cases, an additional infrared sensor is used to provide depth information and to remove the mirage images[25,26]. One major drawback of polarization-stereo-vision methods is that they do not result in correct depth measurements of the reflective surfaces (e.g., the depth of the glass).

Active projectors, such as laser range finders and structured light projector, are widely used in the field of depth sensing. In the case of laser range finder, in order to deal with the reflective surfaces, some researchers use the reflected intensity profile to determine the glass area[27–29]. The fusion of laser range finder and sonar[30,31] is also developed for navigation in the glass environment. In addition, the method used in polarization imaging is applied to the laser range finder, in

[1]School of Information Science and Technology, Fudan University, 200433 Shanghai, China. [2]Berxel Photonics Co. Ltd., 518071 Shenzhen, China. [3]Peng Cheng Laboratory, 518055 Shenzhen, China. [4]These authors contributed equally: Xuanlun Huang, Chenyang Wu. ✉e-mail: connie.chang@berxel.com

which the degree of polarization is calculated and serves as a detection standard of glass areas[32]. However, these methods require scanning around to determine the depth and glass points, thus they are still in the stage of 2D route mapping. As for structured light projectors, a method[33] employs the fusion of Kinect and sonar to obtain the depth images of glass scenes. But, due to the sparse data and narrow angle of the sonar, it needs to sweep multiple times to obtain enough information. Besides, the fusion with sonar will make the system costly, huge and complex.

Deep learning is also a good candidate for the detection of reflective surfaces[34–37], where a network model trained on abundant images of reflective surfaces is employed to mark out the target area. But we find that even the recent works[36,37] might still misjudge the areas with frames or borders, such as an empty frame without glass, as the reflective surfaces.

Therefore, a compact and robust method is still needed for applications in environments full of highly reflective surfaces. Here, we develop a polarization structured light (PSL) 3D sensor[38] with polarization properties on both the transmitter (TX) and receiver (RX). In TX, high-contrast-grating (HCG) vertical-cavity surface-emitting lasers (VCSELs) are specially designed to provide structured light with a strong polarization selection ratio. In RX, a polarization-selection CMOS camera is designed to receive the signal selectively. According to Fresnel's theory[39], the specular reflection from a reflective surface maintains the same polarization as the incident polarized light. However, diffused reflection from objects without smooth surfaces does not exhibit any polarization even incident by a strongly polarized light.

Hence, using a polarization-selection CMOS camera can differentiate reflection from a reflective surface. Thus, the depth information of a reflective surface or objects behind it can be obtained based on the choice of the polarizer direction. Here, we report three experiments to demonstrate how PSL 3D sensors can be used to see as well as to see through highly reflective surfaces.

## Results

### Polarization structured light 3D sensor

The PSL 3D sensor is illustrated in Fig. 1a. It consists of a transmitter TX, a receiver RX and an RGB camera. In TX, we have embedded an HCG-VCSEL array inside, whose working wavelength is 940 nm. VCSELs are critical sources for dot projectors used in structured light cameras[40]. Typical VCSEL uses distributed Bragg reflectors (DBRs) as its top and bottom mirrors. The DBRs are many 10s of layers of planar hetero-epitaxial layers and do not provide or maintain a fixed polarization to the VCSELs. HCG is a thin-film subwavelength metastructure that is effective in providing high reflection with a fixed polarization. Hence, HCG VCSELs have exhibited a very high polarization selection ratio independent of operating temperature or drive conditions[41]. With this specially designed HCG-VCSEL array, the TX of PSL 3D sensor can project dot-array structured light (see Supplementary Fig. S1 and Methods) in either transverse electric (TE) or transverse magnetic (TM) polarization.

The polarization direction of the structured light emitted from TX is denoted by a double-head blue arrow in Fig. 1a and its projection range is 75.5° × 65.7°, which is indicated by a pink area bounded by

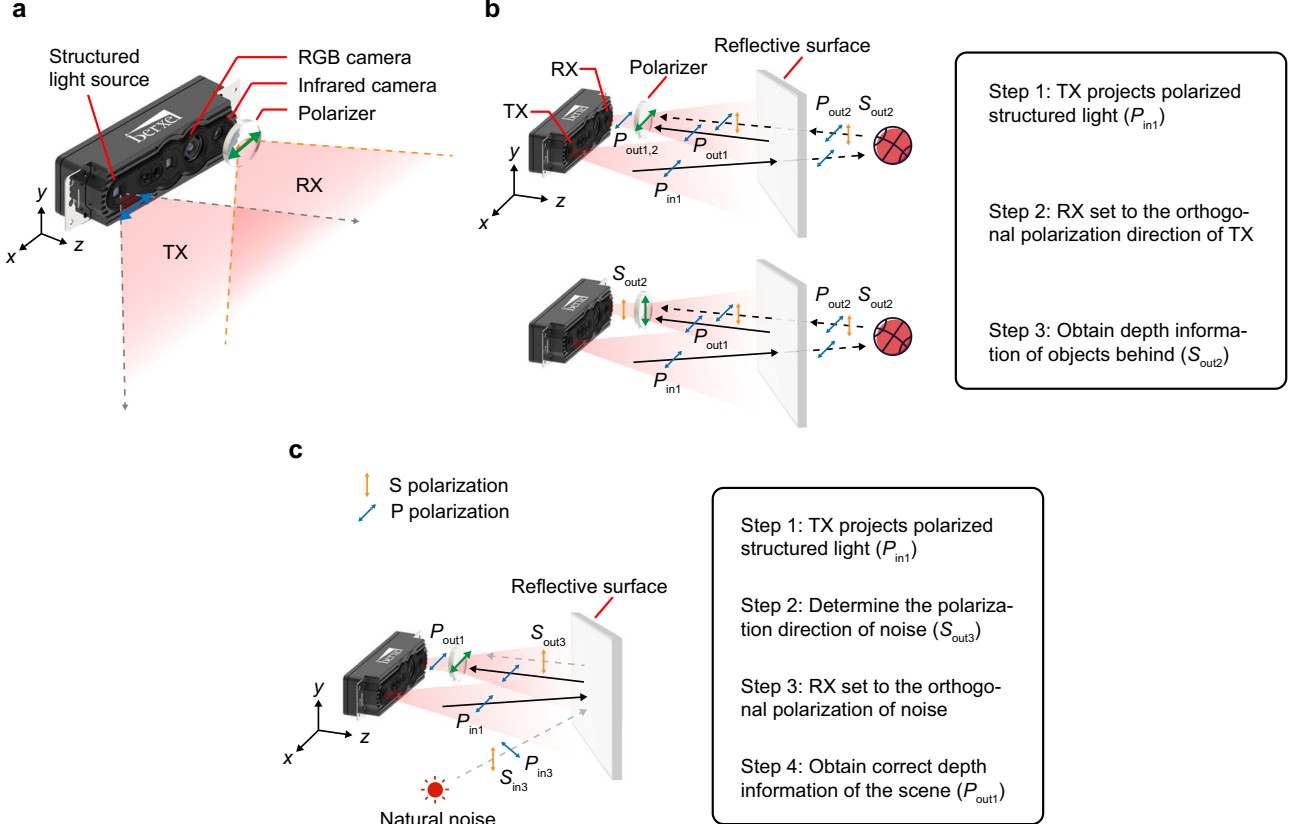

**Fig. 1 | Principle of the polarization structured light 3D sensor. a** The model of PSL 3D sensor. In TX, polarized structured light is produced by an HCG-VCSEL array inside. In RX, signals are received selectively by a rotatable polarizer mounted on the receiving infrared CMOS camera. In addition to TX and RX, PSL 3D sensor contains a RGB camera in the middle as well. **b** The scheme of seeing through the reflective surface. On the upper row, it shows the 3D sensor can obtain depth information of both the reflective surface and the objects behind. On the lower row,

the 3D sensor can remove the depth of the reflective surface. The specific steps are shown in the flow diagram. **c** The scheme of seeing against reflective noise. On the left side, it shows the 3D sensor can obtain the correct depth of the scene by eliminating the effect of reflective noise. The flow of this mission is shown on the right. Note that the plane of the PSL 3D sensor is set to be *x-y* plane and the light propagation direction is in *z* direction.

gray dot arrow lines. Here, we set the plane of the PSL 3D sensor to be $x$-$y$ plane and the light propagation direction to be $z$, then define the horizontal ($x$ direction) polarization, as pointed by the blue arrow in Fig. 1a, to be TM polarization. In RX end, a rotatable polarizer is mounted on the infrared CMOS camera. Its polarization direction is denoted by a double-head green arrow and its receiving angle is $74° × 50.5°$, which is indicated by a pink area bounded by orange dot arrow lines. The polarizer can be rotated to receive the dot patterns selectively. By calculating the spatial displacements of known dot patterns, depth information can be obtained for an entire image at high frame rates.

## Seeing through the reflective surface

According to the polarization feature of the HCG-VCSEL-based PSL 3D sensor, we can fully exploit its merits for 3D imaging in scenarios with large reflective surfaces. Based on Fresnel equations[39], when S- or P-polarized light is incident on a highly reflective surface, the reflection will maintain the same polarized direction. As shown in the upper picture of Fig. 1b, TX of the PSL 3D sensor projects TM-polarization (P-polarization, $P_{in1}$) structured light onto the scene. The reflected light from the reflective surface ($P_{out1}$), which is denoted by the black solid line, will bounce back with the same polarization as the TX, while the light reflected from the object behind the reflective surface ($S_{out2}$, $P_{out2}$), whose light path is denoted by the black dot line, is polarized in various directions due to diffuse reflection. Then RX can receive the signals ($P_{out1,2}$) of both the reflective surface and the object behind with its polarizer set in the same polarized direction, i.e., $x$ direction. As shown in the lower picture of Fig. 1b, if we change the direction of the polarizer to be orthogonal, i.e., $y$ direction, the depth information of the reflective surface is eliminated, leaving that ($S_{out2}$) from the diffuse object behind.

Thus, for seeing through the reflective surface, the process can be divided into three steps, as illustrated in the flow of Fig. 1b. Step 1 is the projection of polarized structured light with a specific polarization direction ($P_{in1}$). Step 2 is the setting of RX polarization direction to be orthogonal to TX. Step 3 is obtaining the depth information of the scene ($S_{out2}$), where the part of the reflective surface ($P_{out1}$) is removed. In addition, the distinction between two polarization settings of RX can be further used to complete the reflective surface.

## Seeing against the reflective noise

In Fig. 1c, we also show the principle of working against reflective noise. In this sketch, the influence of natural light ($S_{in3}$, $P_{in3}$) is indicated by the gray dash line, whose incident plane is in the $x$−$z$ plane. Its reflected component is stronger in S-polarization ($y$ direction, $S_{out3}$) according to Fresnel equations[39]. Hence it can be eliminated when a P-polarized TX and RX are used to enhance the signal-to-noise ratio (SNR), enabling clear 3D image acquisition of the whole scene where there is strong reflective noise from the natural light. The process for this mission can be concluded into four steps, as illustrated in the flow of Fig. 1c. Step 1 is the projection of polarized structured light with specific polarization direction ($P_{in1}$). Step 2 is the determination of polarization direction of noise ($S_{out3}$). Step 3 is the setting of RX polarization direction to be orthogonal to the reflective noise. Step 4 is obtaining the correct depth information of the scene ($P_{out1}$).

## Completing the reflective surface

In addition to seeing through the reflective surface and seeing against the reflective noise, we further present the PSL 3D sensor's capability to detect and complete the reflective surfaces. The two-step working principle is illustrated in Fig. 2. In step 1, we extract the glass region by the collaboration of the depth and color channels. In the depth channel, we first obtain the depth image of the scene in polarization 0°. Polarization 0° means that both the TX and RX of the PSL 3D sensor are in the same polarization, here we set them in TM polarization. Next, we

rotate the RX to the orthogonal polarization, obtaining the depth image in polarization 90°. As illustrated in the principle of seeing through the reflective surface, the glass part can be eliminated because its reflected polarization is orthogonal to the RX. Then, subtraction is applied between these two depth images to capture the area with sharp contrast. Meanwhile, in the color channel, we adopt the deep learning method[36] to get the glass boundary from the RGB image of the scene. For a glass pixel, it is predicted to be 1 (white) and for a non-glass pixel, it is 0 (black). Combining the subtraction result in the depth channel and the boundary result in the color channel, the depth points belonging to the glass can be determined and extracted, which is highlighted in red in the depth image. Note that, for scene without glass, there is no sharp contrast between two polarization settings in the depth channel (see Supplementary Fig. S3 and the results of completing the reflective surface).

After getting the glass region, we move forward to step 2. The extracted glass depth points $P_d = [u, v, z]^\top$ are transformed to the world coordinate of the infrared camera $P_{ir} = [x, y, z]^\top$ using the following equation:

$$
\begin{aligned}
x &= \frac{z(u - c_x)}{f_x} \\
y &= \frac{z(v - c_y)}{f_y},
\end{aligned}
\tag{1}
$$

where $\{f_x, f_y, c_x, c_y\}$ are the internal parameters of the infrared camera. Then the glass is fitted to a large enough area in this world coordinate (Supplementary Fig. S4 explains why it needs to fit in the world coordinate). The fitted glass points are next transformed to the world coordinate of the RGB camera $P_{rgb} = [x', y', z']^\top$ with the external parameters, i.e. the rotational matrix $R$ and translational matrix $T$:

$$
P_{rgb} = RP_{ir} + T.
\tag{2}
$$

These points $P_{rgb}$ are mapped to the color channel $P_c = [u', v', z']^\top$ using equation (1) but with the internal parameters of the RGB camera $\{f'_x, f'_y, c'_x, c'_y\}$. In the color channel, we can interpolate inside the glass boundary according to the fitted glass points $P_c$ and finish the completion of the glass. At last, the completed point cloud can be converted back to the world coordinate. As shown in the right part of step 2, the original point cloud is plotted in the first row, where only the middle region of the glass is detected. After applying step 1 and 2, the whole glass area can be completed, which is highlighted in the second row with red color. With this proposed method, not only the glass can be seen and completed correctly, but also the objects behind the glass can be reconstructed, enabling seeing and seeing through the reflective surfaces.

## Analysis of 3D imaging in scenes with reflective surfaces

Before showing the results of three experiments, we first analyse 3D imaging in scenes with reflective surfaces. Among the reflective surfaces, the indoor glass wall is a typically challenging situation (Fig. 3a, b). In this scene, we place the sensor in front of a glass wall, with a reflecting image of two persons and the sensor on the same side of the glass at about 1.2 m distance in front. In this situation, the depth image obtained with a stereo vision camera (Intel RealSense D455) is shown in Fig. 3c, which is unable to see the depth of the glass. In fact, the stereo vision camera provides erroneous depth measurements based upon the image of the persons reflected by the glass with a measured distance of about 2.4 m behind the glass wall, whereas the actual should be 1.2 m in front of the glass wall. In contrast, when TM polarized structured light patterns produced from the TX of PSL 3D sensor hit the glass, depth information of part of the glass can be reconstructed with TM polarization in RX, as shown by the area encircled by the dashed line in Fig. 3d. These measured values agree well with the actual

**Step 1**

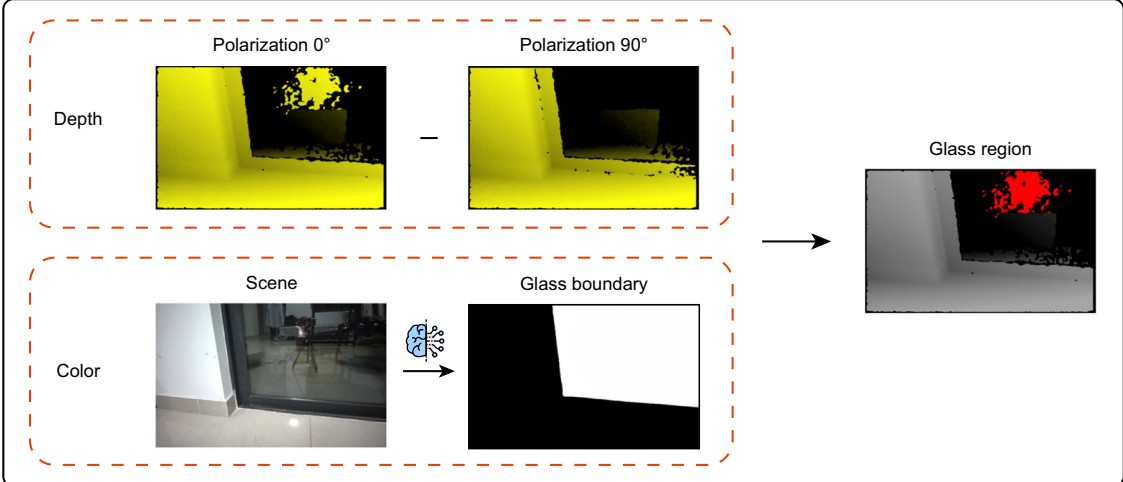

**Step 2**

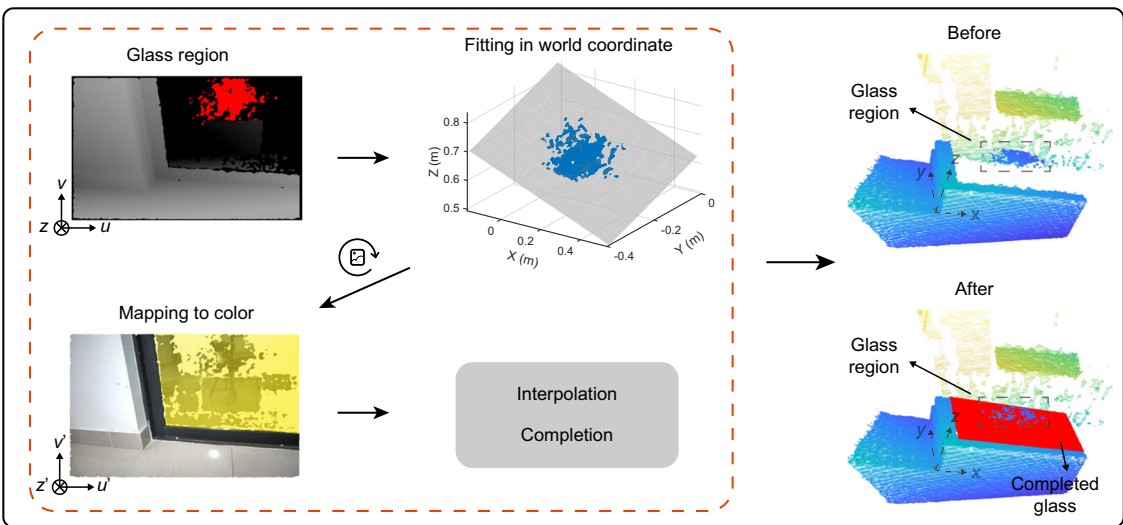

**Fig. 2 | Principle of completing the reflective surface.** In step 1, the glass region is extracted through a combination of depth and color channels, where the glass points are highlighted in red. In depth channel, two depth images are obtained in polarization 0° and 90° respectively. Then, subtraction is employed between these two depth images for extraction of the glass points. In color channel, the glass boundary is predicted from the color image of the scene using deep learning method. In step 2, the extracted glass points are transformed to the world coordinate for fitting. After fitting, the results are mapped to the color channel. In this channel, the reflective surface can be interpolated and completed inside the glass boundary. At last, the completed point cloud can be converted back to the world coordinate, as shown in the before and after comparison on the right side.

distance and this correct depth information can be further used to demonstrate and complete the glass wall.

Deep learning is an effective tool to detect glass[36,37], where a trained deep neural network is used to predict the glass part. However, such RGB-image-based deep learning method will cause misjudgement in some scenes. As shown in Fig. 3e, a cardboard box with its front surface removed is placed in front of the sensor. Inside the box, there is a box of gloves and a metal breadboard. The RGB image of this scene is sent to the network for prediction and its result is shown in Fig. 3f. The prediction result is the probability of being considered as glass, whose range is 0 to 1. For a glass pixel, it is evaluated as 1 (white) and for a non-glass pixel, it is 0 (black). From the result, we can see that almost the whole front surface area is considered to be the glass area. This is because the context of having box boundaries and objects inside will lead to errors in prediction. Moreover, we test another glass door scene. As shown in Fig. 3g, we place the sensor in front of a soundproof room with its door open. That is, there is glass on the left and no glass on the right. Inside the soundproof room, a cardboard box with a basketball on its top is placed on the left, and another box and a book are on the right. Again, the result in Fig. 3h shows that the network predicts both parts as glass areas.

Due to these disadvantages, a much more robust method for the detection of reflective surfaces is required to develop. On the one hand, it needs to obtain the correct 3D information on reflective surfaces. On the other hand, it should be able to determine whether the depth information belongs to them. In the following, we will illustrate how the PSL 3D sensor and its corresponding imaging methods facilitate in such a mission.

## Results of seeing through the reflective surface
We start with a typical situation, i.e. glass door with objects behind, to illustrate PSL 3D sensor's ability to see through the reflective surface. We analyze different factors that can affect this mission, including the distance from the 3D sensor to the glass, the 3D sensor incident angle, the distance of objects behind glass and the density of objects behind glass. Their specific conditions are listed in Table 1.

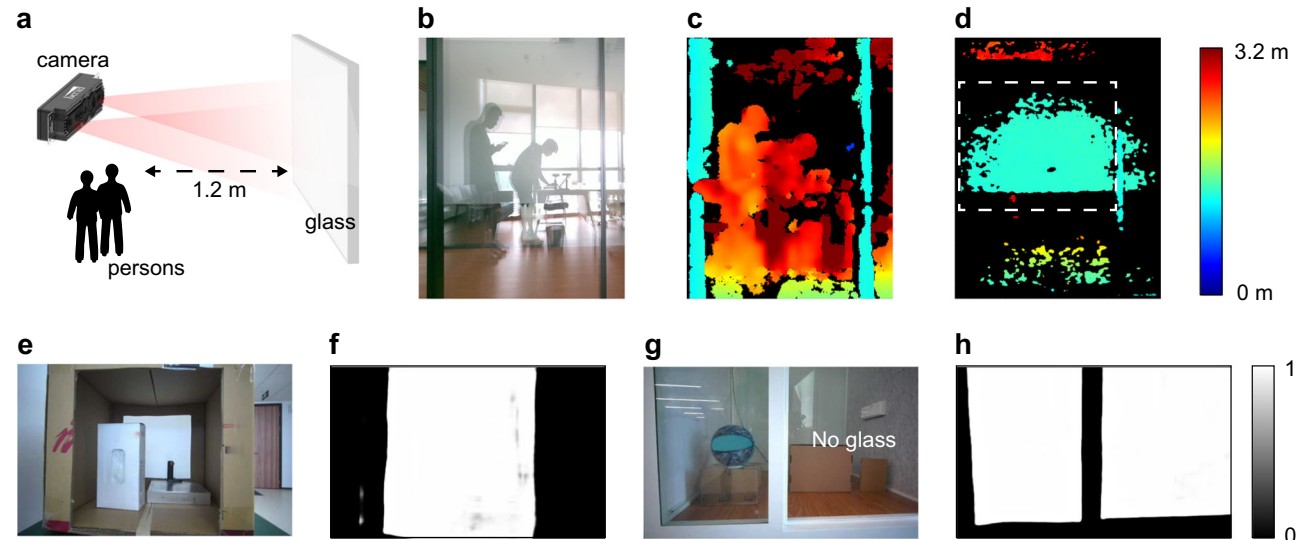

**Fig. 3 | Analysis of 3D imaging in scenes with reflective surfaces.** (**a**) and (**b**) are scheme and picture of the indoor glass wall scene. The glass wall is about 1.2 m in front of the sensor, with a reflecting image of two persons and the sensor. Note that two persons are on the same side as the sensor. **c** Depth image from the stereo vision camera (Intel RealSense D455). **d** Depth image from the PSL 3D sensor. The glass part is denoted by the dashed line. (**e**) and (**g**) are testing scenes of the deep learning method. Their corresponding prediction results are shown in (**f**) and (**h**). The jet color bar applies to (**c**) and (**d**), the gray color bar applies to (**f**) and (**h**).

In Fig. 4, we show the influence of the first factor. The color pictures of the scene are shown in the first column. In this situation, the PSL 3D sensor is facing the glass door with a distance changing from 0.4 m to 1.2 m. Behind the glass door, there is a movable cabinet with a basketball on the top. Their distances to the glass are fixed to 0.5 m. The TX of the PSL 3D sensor projects TM-polarized structured light to the scene. Then the structured light is reflected back to the RX, containing depth information of both the glass door and objects behind. As mentioned in the principle, the reflected light of the glass door maintains the same polarization as the TX, while those of the cabinet and basketball are in various polarized directions due to diffuse reflection.

Hence, when we set the RX to be TM polarization, depth information of the glass door, cabinet, and basketball can all be reconstructed, which is shown in Polarization 0° of Fig. 4a. In this depth map, bright yellow represents the near points and black represents the far or missing points. The middle bright yellow area accurately reveals the depth of the glass, while the farther cabinet and basketball are darker. Unlike Fig. 3d, the central part shows the depth of the basketball rather than the glass because reflection from the basketball is stronger. Next, if we rotate the RX to the orthogonal direction, i.e. TE polarization, the glass part can be filtered, leaving clear depth information of the cabinet and basketball (Polarization 90° in Fig. 4a). As we change the distance to 0.8 m, the middle ball area becomes smaller because the glass depth begins to dominate (dot line in Polarization 0° of Fig. 4b). As for 1.2 m (Fig. 4c), the middle is almost replaced by glass depth, while the whole proportion of glass depth is smaller as compared to that of 0.4 m. In Polarization 90° for both cases, the glass part can still be eliminated.

More details about the incident angle of 3D sensor, distance and density of objects behind glass are displayed in Supplementary Fig. S5–S8. Regardless of different incidence angles, or different distances and object densities, the 3D sensor can eliminate the depth of the glass and achieve the function of seeing through the reflective surface. Supplementary Movie 1 clearly shows this phenomenon by rotating the RX polarizer 360 degrees. The high contrast between Polarization 0° and Polarization 90° can further serve as a determination method of the reflective surface, which will be demonstrated in the section after next.

### Results of seeing against the reflective noise

We also set up outdoor and indoor scenes to demonstrate the advantage of detecting against noise (Fig. 5). We first show a case of outdoor glass scene (Fig. 5a). In this situation, a book is placed behind a tilted glass, where there is strong specular reflection from the natural light. We can see that the reflected noise incapacitates the stereo vision camera for obtaining the depth of the scene, leading to black-irregular-patch errors in the depth map (encircled by white and red dot lines while the book is indicated in red). However, as mentioned in the Fig. 1c, this noise has a larger component of S polarization. Note that in this setting, S polarization is in the horizontal direction ($x$ direction) when facing the glass. Therefore, we can utilize P-polarized (i.e. $y$-direction polarization) TX and RX to filter this noise, enhancing the SNR. As shown in Fig. 5c, the book behind the glass (encircled by the red dot line) can be measured clearly with the PSL 3D sensor in this P-polarized setting.

Besides this sunlight noise, the shadow of objects may also severely blind the depth camera based on stereo vision. As illustrated before in Fig. 3c, the reflection noise of two persons makes the stereo vision camera provide erroneous measurement. Another outdoor scene with reflection overlapped with the detecting object is also illustrated in Supplementary Fig. S9, where the stereo vision camera is still affected by the reflection of outside objects while the PSL 3D sensor can get the depth information correctly.

In the next case, we analyze the ability to detect against noise in the situation of wall corner, which is a common scene for indoor serving robot. The results are displayed in Fig. 5d–l. As shown in Fig. 5d, the comparison is first done at a height of 0.24 m and a distance of 0.7

### Table 1 | Influence factors for seeing through the reflective surfaces

| Influence factor | Specific experimental condition | | |
|---|---|---|---|
| Distance from 3D sensor to glass | 0.4 m | 0.8 m | 1.2 m |
| 3D sensor incident angle | −30° | 0° | 30° |
| Distance of objects behind glass | 0.3 m | 0.5 m | 0.7 m |
| Density of objects behind glass | Low | Medium | High |

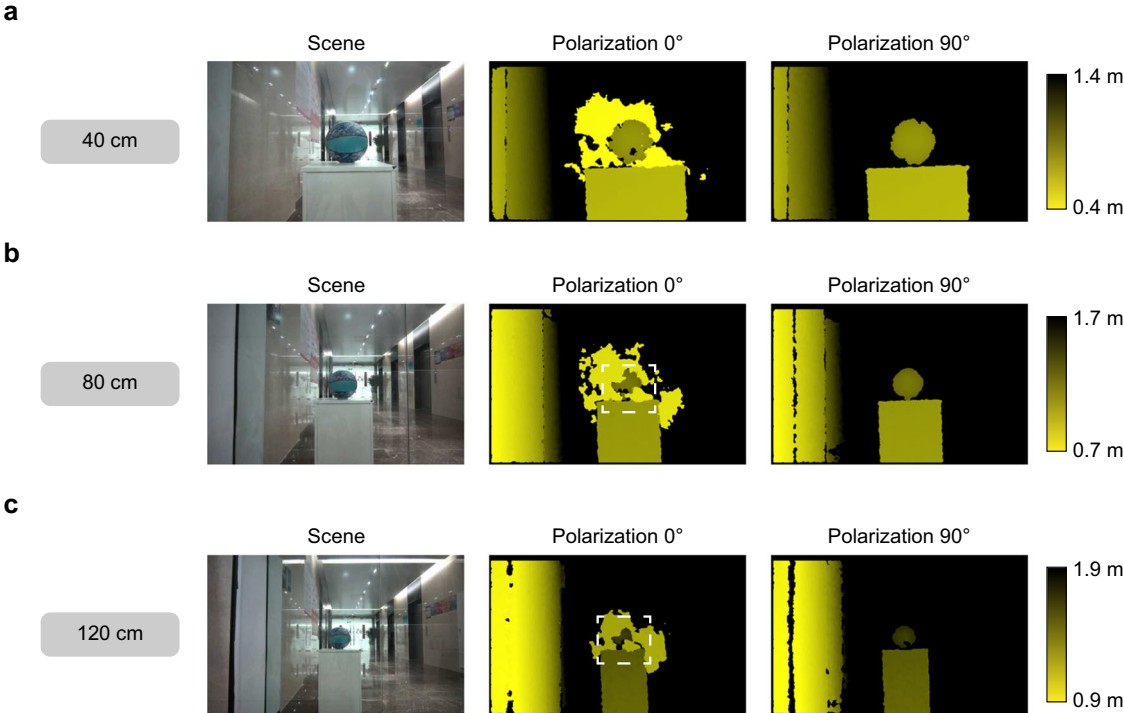

**Fig. 4 | Seeing through the reflective surface. (a)–(c)** are the results of different distances from 3D sensor to glass. Polarization 0° is the depth map when TX and RX are in the same polarization. Polarization 90° is the depth map when TX and RX are in the orthogonal polarization. The yellow-black color bars apply to both second and third columns.

m away from the corner, with the sensor 45° downward. In this situation, due to the influence of reflection and multipath noise, the stereo vision camera will make the wrong depth judgment. As indicated by the red dot line in Fig. 5e, the depth has gone deep below the floor plane. Only a small part (orange dot line) has accurate floor information. However, for the PSL 3D sensor (Fig. 5f), a complete and correct shape of the corner can be obtained by setting TX and RX to be P polarization ($y$ direction). Then we change to the height of 0.4 m (Fig. 5g), with a distance 0.7 m and an incline angle 45° unchanged. It means that the incident angle decreases. In this condition, the correct floor area obtained from the stereo vision camera starts to increase (orange dot line in Fig. 5h), but the depth area encircled by the red dot line still penetrates into the floor. As we change to the height of 0.56 m (Fig. 5j), the floor area of the stereo vision camera goes back to the correct plane (Fig. 5k), but there are still some black speckles in the depth map because of noise. However, for the PSL 3D sensor, the intact depth map and point cloud are maintained in these two heights (Fig. 5i and l) with this P-polarized setting.

For these common scenes of reflective surface, PSL 3D sensor can remove noise through specific polarization combinations, improve the signal-to-noise ratio, and obtain complete depth information of the scene. More polarization combination experiments and comparison with time of flight (ToF) camera, presented in Supplementary Fig. S10–S13, further demonstrate the superiority of PSL 3D sensor in these scenarios.

### Results of completing the reflective surface
In the following, we prove the feasibility of completing the reflective surfaces. The results of eight common scenes are selected to illustrate in Fig. 6 and Supplementary Fig. S14. In Fig. 6, we display four of these scenes, including balcony glass, soundproof room glass, spherical glass and office door glass. In the balcony glass scene, a yellow square box is placed outside the glass door which has reflections of the interior furniture. With polarization 0° and polarization 90° settings,

we obtain two depth images of the scene. In polarization 0°, partial of the glass and the box outside can both be reconstructed while the glass part is eliminated in polarization 90°. Compared to the stereo vision camera, which fails to detect the glass depth and misjudges the reflections as real objects, such a change in depth channel can serve as a reliable cue for determining the reflective surfaces. Thus, using the subtraction of two depth images and the glass boundary predicted from the RGB image of the scene, we are able to extract the glass region, which is highlighted in red in the fifth row. The extracted glass points are then used to fit and complete the reflective surface. The completed depth image is shown in the seventh row, where the partially empty glass has now been filled in. The final comparison between the original point cloud and the completed point cloud is shown in the last two rows. In the completed point cloud, we can see that the new glass plane, which is colored in red, matches well with the glass frame and the 3D information of the box outside the glass can be acquired simultaneously.

As for the soundproof room glass scene, we execute the same process. In this scene, the right door of the soundproof room is open, that is, only the left side has glass. Inside the room, a cardboard box with a basketball on its top is placed on the left, and another box and a book are on the right. As seen in the depth images, only the left side with glass has a large change while the right side shows no significant change. If simply relying on the predicted glass boundary from deep learning, we will mistakenly consider the right side as glass. However, our method can determine the regions that actually have glass and extract effective depth information belonging to the glass. Similarly, this accurately extracted depth information is used for the following fitting and completion. In the completed results, the left glass door and objects inside the room can be reconstructed, as if exploring the inside and outside of the soundproof room from God's perspective.

Besides planar glass, our method can also be applied to curved glass, which is displayed in the third column. In this situation, we use a spherical exhibition stand with two 3D-printed models inside as the

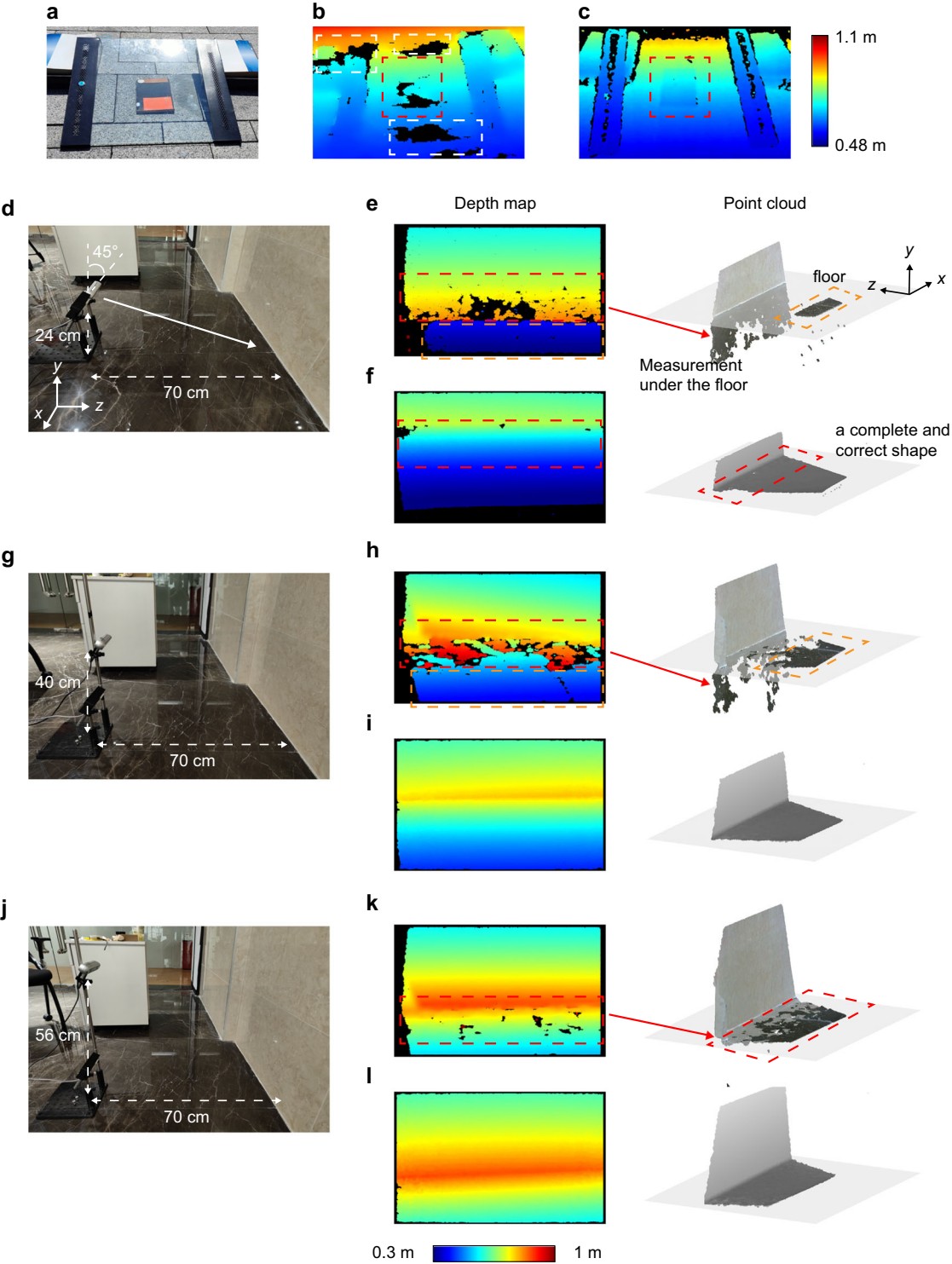

**Fig. 5 | Seeing against the reflective noise.** (**a**)–(**c**) are comparison results of an outdoor glass scene, where (**a**) is the picture of the scene, (**b**) is the depth map from stereo vision camera (Intel RealSense D455) and (**c**) is the depth map from PSL 3D sensor. (**d**)–(**l**) are comparison results of an indoor corner scene. The experiments are done in different height, whose schemes are shown in (**d**), (**g**), and (**j**). The corresponding depth map and point cloud from stereo vision camera are displayed in (**e**), (**h**) and (**k**) respectively, while those from PSL 3D sensor are in (**f**), (**i**), and (**l**). The jet color bar with range (0.48 m, 1.1 m) applies to (**b**), (**c**), and the one with range (0.3 m, 1 m) applies to the depth map column.

detecting object. Likewise, two depth images are obtained from two polarization directions. We can see that the front face of the glass ball is eliminated in polarization 90°. With the ball-shaped glass boundary, we acquire the glass region in the fifth row. These points are from the spherical glass so the fitting result is a quadric surface. Then we complete the glass inside the glass boundary and get the completion result. As seen from the point cloud in the last row, it completes well the front surface of the spherical glass. Note that, there are still some misjudged points in the predicted glass boundary, such as the unsmoothness and the false positive outliers at the ball edge. These minor flaws can be solved by optimizing the predicting network in future work.

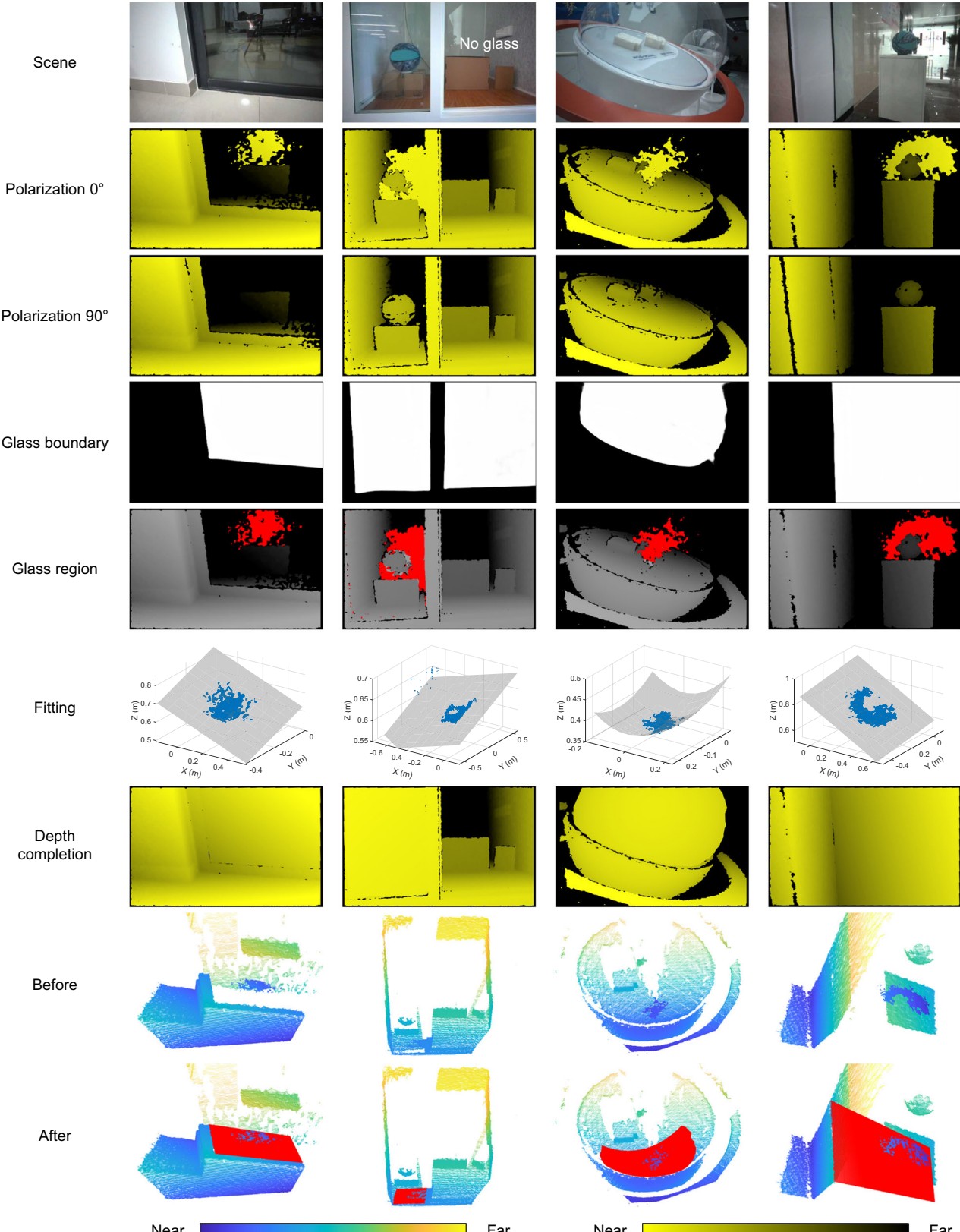

**Fig. 6 | Detection and completion results.** The results of four scenes are presented column-wise in this figure. The first row is the RGB image of each scene. The second and third rows are the depth map from polarization 0° and polarization 90° settings respectively. The fourth row is the predicted glass boundary using the deep learning method. The fifth row is the extracted glass region, which is obtained from the combination of depth map subtraction and glass boundary. The sixth row shows the fitting of the extracted glass points in the world coordinate. The seventh row is the completed depth map. The original point cloud and the completed point cloud are shown in the last two rows. In the completed point cloud, the completed glass is highlighted in red. The parula color bar applies to the last two rows and the yellow-black color bar applies to the rows of polarization 0°, polarization 90°, and depth completion.

At last, we show an office door glass scene with a larger tilted angle than the first column. In like manner, there is a cabinet with a ball on its top behind the glass. Due to different imaging angles and distances, the position of the glass depth is different in polarization 0°, whereas it will not affect the elimination of glass in polarization 90°. With the same subtraction method, the glass points are extracted and shown in the fifth row. Then the glass door is fitted and completed. From the completion results in the seventh row and the last row, we can see that the glass can be completed well and the box and ball behind the glass can also be reconstructed, reinforcing the idea of seeing and seeing through the reflective surfaces.

It can be seen from the above experiments that the position and proportion of effective glass information are variational for different experiment situations. This is because of the combined effect of the illumination angle and the cleanliness of the reflective surface. In Supplementary Fig. S15, we analyze the influence of different cleanliness of the reflective surface on the detection area. In fact, if the reflective surface perfectly follows the specular reflection model, only an area with the size of the emitter will be fully received. But in common scenes, this kind of situation rarely exists on the reflective surface. Actually, it will be affected by the randomly distributed particles on the surface, expanding the reflection range to a certain extent, which is the so-called off-specular reflection[42,43]. As our experiments show, the PSL 3D sensor can extract enough effective glass information in various scenes.

We also describe the effective measurement angles in Supplementary Fig. S16. We fix the position of the sensor facing the glass and then measure the working range by rotating the sensor horizontally and vertically to measure the proportion of the glass points. We use the number at 0° as a base and the ratio down to 2% as the limit, and the measured working range is ±45° horizontally and ±30° vertically. Moreover, the rest of the eight scenes are also illustrated in Supplementary Fig. S14, demonstrating the general applicability of our method.

As for the measurement accuracy, we use the absolute error to evaluate, and the results are shown in Fig. 7. Among them, Fig. 7a-h are the error maps of eight scenes, where Fig. 7a-d correspond to the four scenes in Fig. 6 and Fig. 7e-h correspond to the four scenes in Supplementary Fig. S14. Specifically, we cover the reflective surfaces with thin stickers and conduct depth detection on the scene again, the fitted depth of which is used as a benchmark. We then calculate the absolute error between the depth map of the completed reflective surface and this benchmark. Then we perform boxplot statistics on these eight error maps, and the results are shown in Fig. 7i. The green box includes the median and the upper and lower quartiles of each error map, the gray points are outliers exceeding 1.5 times the interquartile range, and the yellow circle dashed line is the mean value of each error map. From the error analysis, we can see that the errors are maintained at the level of 1mm, demonstrating the precise reconstruction of our method. Note that there are outliers in the error maps.

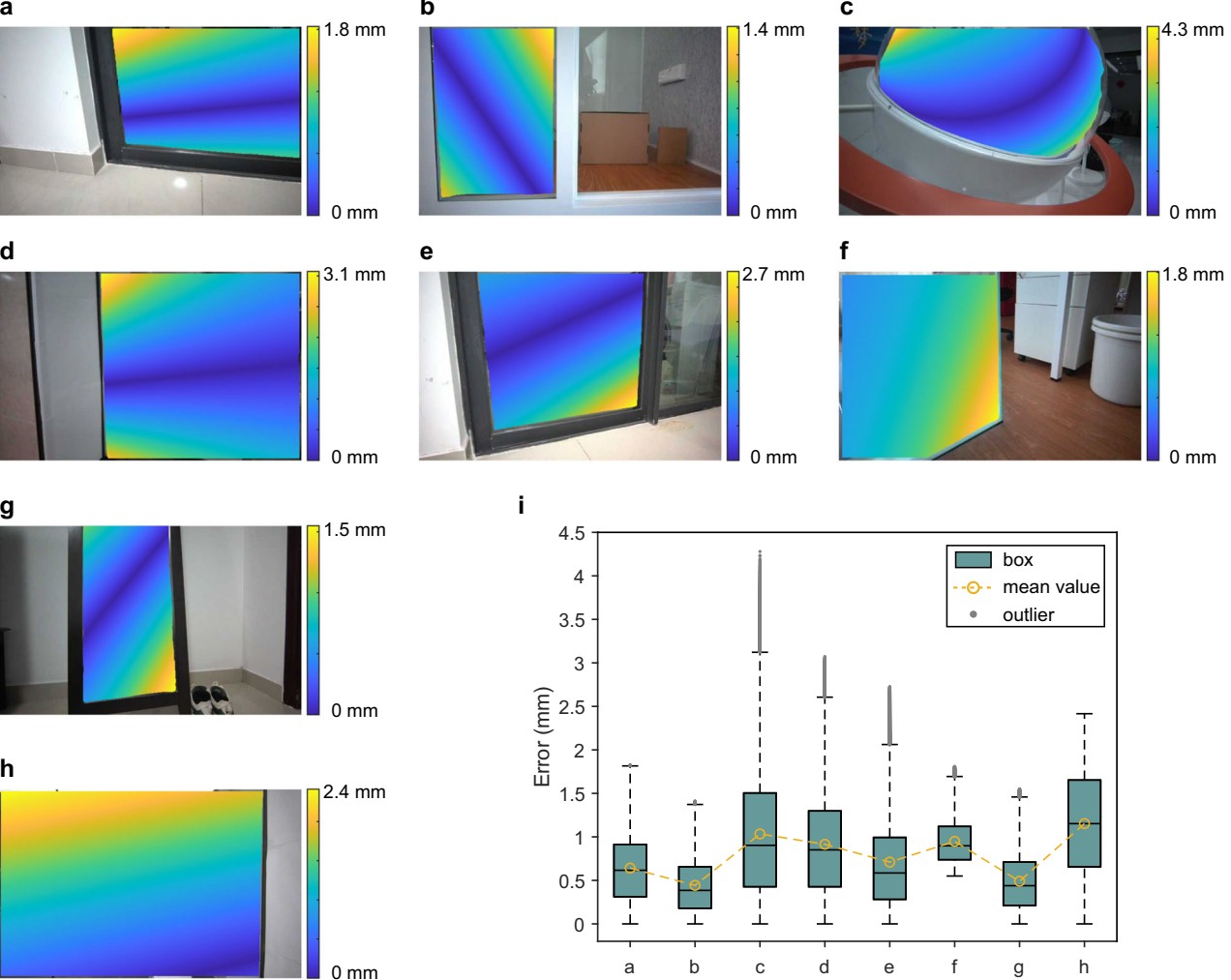

**Fig. 7 | Error analysis.** (**a**)–(**h**) are the error maps of eight scenes of reflective surfaces, which are the absolute errors between the completed reflective surface and the fitted benchmark. **i** The boxplot graph of these eight error maps. It includes the box, mean value and outliers of each map, showing their statistical properties.

These errors mainly come from the fluctuation of the data, such as the influence of the interreflection, which can be reduced to a much lower level if we include the model to suppress it in the future.

The above experiments of completion are from a single view. If we want to reconstruct the whole scene, we need to employ our method in different views. In Supplementary Fig. S17, we show the reconstruction of a fish tank by employing our method in four sides of the fish tank. In addition, a more complicated scene with a mirror at the back of glass is used to prove the feasibility of our method in Supplementary Fig. S18. The ToF camera is also compared in Supplementary Fig. S19, which once again demonstrates the distinction of the PSL 3D sensor.

## Discussion

In summary, we have invented an HCG-VCSEL-based polarization structured light (PSL) 3D sensor and proposed the corresponding imaging methods for 3D reconstruction in a wide type of scenes with highly reflective surfaces.

We first chose a glass scene to demonstrate the ability of the PSL 3D sensor to obtain 3D information of objects behind reflective surfaces. Next, we compare with the stereo vision camera and ToF camera, and show that in the scene with strong reflective noise, through a specific polarization setting, our sensor can eliminate the reflective noise and accurately obtain the 3D image of the scene. Then, we select 8 different scenes to prove that our method can be robustly applied to the detection and completion of the reflective surfaces. Moreover, we show that the effective working range of incident angle is as large as $90° \times 60°$. In contrast, stereo vision cameras, ToF cameras, and deep learning methods will cause erroneous measurements in these scenarios. At last, we evaluate the absolute errors of our detecting results, and the mean error of each scene maintains at the level as low as 1mm, equivalent to 0.1%. Through these different experiments, we demonstrate that the PSL 3D sensor can realize seeing and seeing through the reflective surfaces.

Currently, the proposed methods are conducted in post-processing, but in fact they can be integrated into the module. We can also modify and optimize the design of the deep learning network, so that they can be transferred to the processing chip of the sensor for real-time detection and completion. On the receiving end, we can also design a unique polarizer array like that in[44], which can be coated on the CMOS camera and receive the signal from different polarizations simultaneously, making our system more compact.

Serving as the eyes of a robot, PSL 3D sensors will find many applications such as service robots and logistics robots. These robots will inevitably encounter the problem of reflective surfaces such as the glass door and floor corner shown in the main text. The PSL 3D sensor can make them handle these cases correctly. Besides, object or scene reconstruction of reflective surfaces, such as office and exhibition hall, will also benefit from this method. Thus, with the special polarization characteristic, the PSL 3D sensor can be further extended to a wide range of indoor and outdoor applications.

## Methods
### Generation of dot-array structured light
The polarized dot-array structured light is generated directly from the TX. Inside the TX, there is a 940 nm HCG-VCSEL array, a collimating lens with 4.8 mm focal length and a diffractive optical element (DOE). The HCG-VCSEL array and the DOE are mounted at the focal points of both sides of the lens. By designing the bandwidth and airgap of the HCG, the VCSEL sources produce polarized Gaussian beams. These beams propagate to the lens and are collimated to compensate the divergence. Then the collimated light travels to the DOE. Passing through the DOE, one beam can diffract to $11 \times 9$ orders uniformly. If we properly position the VCSEL on the array, then the far-field distribution of the laser array can be duplicated in $11 \times 9$ copies, enabling the production of dot-array structured light over 30,000 points. The far-field dot-array distribution is shown in Supplementary Fig. S2.

### Postprocessing of the point cloud data and RGB images
The reading and drawing of point clouds are done by the Point Cloud Processing of Computer Vision Toolbox in MATLAB. The fitting uses the fit function of Curve Fitting Toolbox with its robust option set to bisquare. The other processing of the point clouds is also based on MATLAB. The prediction of the reflective surface boundary is based on PyTorch. In addition, the visualization of the point clouds in Fig. 5, Supplementary Fig. S10–S13 and Supplementary Fig. S17 is based on CloudCompare[45].

### Calculation of the depth of reflective surfaces
By calculating the spatial displacements of known dot patterns, depth information of reflective surfaces can be obtained according to triangulation. The received dot patterns on the reflective surfaces are shown in Supplementary Fig. S20 and a model of triangulation is also shown to illustrate how depth is calculated.

## Data availability
All the data and codes that support the findings of this study are available in Supplementary files and Figshare under accession code https://doi.org/10.6084/m9.figshare.23722416[46].

## Code availability
All custom codes are uploaded along with the data, as stated in Data Availability.

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

## Acknowledgements

This research was partially funded by the Shenzhen Science and Technology Programs No. KQTD20200820113053102 and No. ZDSYS20220325163600001; the Guangdong Major Talent Introduction Project (No. 2021ZT09X328); the National Key Research and Development Program of China (2022YFB2802803); the Natural Science Foundation of China Project (No. 61925104, No. 62031011) and Major Key Project of PCL.

## Author contributions

C.J.C.-H. conceived and supervised the project. X.H. wrote the processing codes. X.H. and C.W. performed the experiments and analyzed the data. X.X., B.W., S.Z., and C.Y. designed, assembled, and tested the PSL 3D sensor. C.S. and J.W. designed and managed the fabrication of HCG VCSELs. N.C. and S.Y. provided guidance and advice on experiments. X.H., C.W., and C.J.C.-H. interpreted the data and wrote the paper, with input from all the other authors.

## Competing interests

The authors declare no competing interests.
