## [Peer Review File · Nature Communications]

Polarization Structured Light 3D Depth Image Sensor for Scenes with Reflective SurfacesReviewer #1 (Remarks to the Author):

The paper is a worthwhile contribution to 3D depth imaging technology using existing ideas from polarisation imaging. While definitely publishable, the paper needs a lot of work in terms of structure and experimentation. The paper seems to jump between results and method, making it very hard to follow in places. Replace section "2. Results" with "2. Method" and clearly outline the approach one step at a time. Perhaps add a flow diagram or pseudocode. Then add a new section to describe the results. The experiments need to be more systematic. E.g. start with a sphere placed behind a glass pane and analyse accuracy. Then gradually change the angle of incidence. Also, a more systematic approach to analyse the affect of background noise would be beneficial. All this could replace the HUGE amount of text (even the most serious researcher would struggle to keep awake with so many words in the second half of the paper!).

A few minor points:

- The term "reflection surfaces" is odd and not well known. I wonder if "highly reflective surface" might be better.
- Intro mentions that refractive index needs to be known for existing methods but this is not always the case.
- Intro sentence starting "However, because of..." is not clear.
- Intro "surrounded by some borders" also not clear. There are quite a few sentences like this that will not make sense to most readers.
- Define "polarization selection ratio".
- Another couple of sentences to explain the core principle of your method is needed in the intro.
- The captions are huge and repeat much text from the main body. They should be much shorter.
- In section 2, it is clear about how the reflective surfaces are detected. But more information needed on how depth is calculated.
- The effect of natural noise depends on incidence angle. This is glossed over.
- I'm confused about the geometry considered for fig 2. Maybe add a diagram?

Reviewer #2 (Remarks to the Author):

Polarization imaging has been studied and applied for a long time, always with a polarizer after the source to generate the polarized light. The paper entitled "Polarization Structured Light 3D Depth Image Sensor for Scenes with Reflection Surfaces" presents a 3D sensor with polarized structured light for reflection surfaces and shows several examples of seeing glass surfaces. The PSL camera includes an HCG-VCSEL array with an intrinsic high polarization selection ratio as the laser source without a discrete polarizer. The results in the paper are very interesting, and show promising applications. This work will stimulate the research of polarization imaging in 3D sensing. I have the following comments and questions.

1. In supplementary Fig. S4, it is better to give a schematics to show "measure the working range of the sensor by rotating it in the horizontal and vertical directions". It is not easy to understand the measurement to the readers.
2. In the last row of Fig. 5, What does the red region present? "In the completed point cloud, the glass is displayed in red.", I can't understand why the authors add a red region in a tilted style in the point cloud.
3. In this polarization imaging application, one challenge is the polarized laser source. Previously, a polarizer is used after the laser source to generate the polarized light. One novelty in this work is using the HCG-VCSEL array with a high polarization selection ratio as the laser source, eliminating the discrete polarizer. In the supplements, two papers are cited to show the spectral high polarization selection ratio of 30 dB. Still, I think the power curve and the power OPSR curve of the HCG-VCSEL array should be included in the supplement. Since the thermal crosstalk exists in the array, is the power OPSR of the HCG-VCSEL array the same as that of the single HCG-VCSEL? Can the authors analyze the influences of the polarization ratio on the image quality of the PSL camera? The power of the VCSEL array in the 3D sensing can affect the depth accuracy. How about the output power of the HCG-VCSEL array used in the experiments?

4. In Fig. 5, Column 2-4, Row 8, some point cloud regions are missing. What are the reasons? How to solve this issue?
5. The PSL camera shows the advantage to see the reflection surface. I see in the paper that, the error can reach 80 mm, and the average error of the entire plane is at the level of 10 mm. As far as I know, the current structured light 3D camera can reach an accuracy of 1 mm at 1-2 m distance. The results seems that the PSL camera has no competition in the depth accuracy, compared with the current structured light camera for 3D sensing. How to achieve an accuracy of 1 mm at 1-m distance with the PSL camera?
6. In the conclusion section, I suggest to give some discussions on the potential application fields of the PSL camera.
7. This paper presents the results of PSL camera with HCG-VCSEL array. Another approach of 3D sensing is time of flight (TOF) method. I suggest that the authors give some discussions on the TOF polarized imaging with HCG-VCSEL array.

Reviewer #3 (Remarks to the Author):

1. It can be seen from the method part of the paper that the author's understanding and use of polarization optics is relatively simple, so it is suggested to learn relevant knowledge;
2. As a device with similar functions to kinect, there have been relevant studies similar to this paper since 2009. Please carefully investigate relevant literatures;
3. In the past 20 years, there has been a lot of work using polarization optics to separate specular reflected light and carry out depth estimation. The algorithm and hardware equipment in this paper have no innovation, and the separation effect is not good, which has no advantage compared with relevant achievements after 2010;
4. This paper uses the form of rotating polarizer to acquire polarized light, which is a time sequence acquisition method, its application scenario is greatly limited, and its practical application value is low;
5. There is no literature related to polarized light separation specular reflection component in the references, which reflects that the author has no knowledge accumulation in this field and does not understand the latest relevant research results;
6. The paper lacks the introduction of mathematical model and corresponding schematic diagram (but the model of pol0-pol90 is too simple, so the reviewer does not expect the summary diagram of this model);
7. In the noise experiment to remove the reflection of natural light, what should be done if the reflected noise area overlaps or partially overlaps with the measured target? In the experiment of this paper, the two are separated from each other, which is a special and simple case. The author did not solve the key problems mentioned above.
8. Removing specular reflection by rotating the polarizer is not accurate. Specular reflection is local information, which is related to incident light Angle, etc. The difference between two images cannot guarantee the detection and removal of reflection area. If there is a strongly reflective object behind the glass, the device will fail.

Response to Reviewers' Report: NCOMMS-23-10384

We appreciate the editors' and reviewers' time and efforts in reviewing our manuscript entitled "Polarization Structured Light 3D Depth Image Sensor for Scenes with Reflective Surfaces". According to the suggestions and comments of the reviewers, we have substantially revised our manuscript.

In the following, we outline the reviewer's comments (in **black bold**) and the responses to those specific comments (in **blue**) as well as revisions to the paper based upon these comments (**red**). We hope the reviewers will be satisfied with our revision.

Response to the Report of Reviewer #1:

The paper is a worthwhile contribution to 3D depth imaging technology using existing ideas from polarisation imaging. While definitely publishable, the paper needs a lot of work in terms of structure and experimentation. The paper seems to jump between results and method, making it very hard to follow in places. Replace section "2. Results" with "2. Method" and clearly outline the approach one step at a time. Perhaps add a flow diagram or pseudocode. Then add a new section to describe the results.

We thank the reviewer for the confirmation of our work and the helpful advice on the revision. According to the suggestions on the structure of the manuscript, we have reorganized the "2. Principles" section to include the composition of PSL 3D sensor and the working principles of the three experiments: seeing through the reflective surface, seeing against the reflective noise, completing the reflective surface.

The specific subsections are entitled: "Polarization structured light 3D sensor", "Seeing through the reflective surface", "Seeing against the reflective noise" and "Completing the reflective surface". We clearly describe our approaches first in these subsections. The corresponding Fig. 1 and Fig. 2 are also refined in terms of diagram and caption to support the descriptions.

In addition, their corresponding experimental results are described in the "3. Results" sections with subheadings "Results of seeing through the reflective surface", "Results of seeing against the reflective noise" and "Results of completing the reflective surface".

The experiments need to be more systematic. E.g. start with a sphere placed behind a glass pane and analyse accuracy. Then gradually change the angle of incidence. Also, a more systematic approach to analyse the effect of background noise would be beneficial. All this could replace the HUGE amount of text (even the most serious researcher would struggle to keep awake with so many words in the second half of the paper!).

We appreciate the reviewer's advice on the experiments. We have added two new parts of experiments (Fig. 4 and Fig. 5) in the main text and ten more pages in the Supplementary Information to systematically analyze the ability of our PSL 3D sensor.

To be specific:

Firstly, in the experiment of seeing through the reflective surface, a new table of influence factors (Table 1) is used to prove that regardless of different incident angles, or different distances and object densities, our PSL 3D sensor can eliminate the depth of the glass and achieve the function of seeing through the reflective surface. The new results are included in Fig. 4 of the main text and Supplementary Section V. And the corresponding modifications in the main text are the whole section of “Results of seeing through the reflective surface ...” (Page 5-6).

A Supplementary Movie is also added to show the elimination effect by rotating the RX polarizer 360 degrees. In addition, the comparison with the common DBR-VCSEL-based sensor, which is unable to achieve the elimination function, is added in the last paragraph of Supplementary Section V “Meanwhile, ...”, showing the distinction of our PSL 3D sensor.

Table 1: Influence factors for seeing through the reflective surfaces

Influence factor	Specific experimental condition		
Distance from 3D sensor to glass	0.4m	0.8m	1.2m
3D sensor incident angle	-30°	0°	30°
Distance of objects behind glass	0.3m	0.5m	0.7m
Density of objects behind glass	low	medium	high

Secondly, in the experiment of seeing against the reflective noise, comparisons with stereo camera and ToF camera in different scenes and different heights (incident angles) are carried out to prove the advantages of our PSL 3D sensor.

In the case of glass scene, two kinds of outdoor situations (Fig. 5a-5c and Supplementary Fig. S9a-S9c) are used to show that the reflective noise, like sunlight and shadow of objects, incapacitates the stereo vision camera for obtaining the depth of the scene, while the PSL 3D sensor can measure them clearly. The corresponding main text contents are the first and second paragraphs of the section “Results of seeing against the reflective noise” (Page 6, “Results of ... Besides this sunlight noise, ... while PSL 3D camera can get the depth information correctly.”)

In the case of wall corner scene, comparisons in different heights (i.e., different incident angles) with stereo vision camera and ToF camera (Fig. 5d-5l and Supplementary Fig. S13) are added to show that our PSL 3D sensor can obtain the correct depth information with its P-polarized setting at all heights while the other two cameras get the wrong depth measurements due to the reflection and multipath noise. The corresponding main text content is the third paragraph of the section “Results of seeing against the reflective noise” (Page 6-8, “In the next case, ... with this P-polarized setting”)

Moreover, experiments of polarization combinations are added in the paragraphs 2-4 of Supplementary Section VI (Supplementary Fig. S10-S12) to show the advantages of

the PSL 3D sensor. Through all these common scenes of reflective surfaces, our PSL 3D sensor is proved that it can filter noise by specific polarization combinations, improve the signal-to-noise ratio and obtain complete depth information of the scene.

Figure 5: **Seeing against the reflective noise.** (a)-(c) are comparison results of an outdoor glass scene, where (a) is the picture of the scene, (b) is the depth map from stereo vision camera (Intel RealSense D455) and (c) is the depth map from PSL 3D sensor. (d)-(l) are comparison results of an indoor corner scene. The experiments are done in different height, whose schemes are shown in (d), (g) and (j). The corresponding depth map and point cloud from stereo vision camera are displayed in (e), (h) and (k) respectively, while those from PSL 3D sensor are in (f), (i) and (l). The jet colorbar with range (0.48m, 1.1m) applies to (b), (c), and the one with range (0.3m, 1m) applies to the depth map column.

Figure S9: **Comparison on an outdoor glass scene.** (a) Scene of the experiment. (b) Depth map from the stereo vision camera. (c) Depth map from the PSL 3D sensor.

Figure S13: **Detecting results from ToF sensor.** (a)-(c) are depth maps at different heights. (d)-(f) are their corresponding point clouds. The jet colorbar applies to (a)-(c).

Thirdly, in the experiment of completing the reflective surface, all the previous scenes (Fig. 6, Fig. 7 and Supplementary Fig. S14) are remeasured to achieve high precision at the level of 1mm. In addition, new experiments about the fusion of a fish tank, the detection in the complicated scene where a mirror is placed behind the glass and the comparisons with the ToF camera are new included in Supplementary Section X-XII (Supplementary Fig. S17-S19) to verify our method.

A few minor points:

- The term “reflection surfaces” is odd and not well known. I wonder if “highly reflective surface” might be better.

Thanks for the advice. We have changed all the terms to (highly) reflective surface.

- Intro mentions that refractive index needs to be known for existing methods but this is not always the case.

We have revised the way of this description and added three papers that ease the need of refractive index. The modification is highlighted in the second paragraph of Introduction (Page 1):

“However, polarization imaging needs to capture multiple polarized images and can only estimate relative depths. In addition, most of them requires knowledge of the refractive index of the object, while some rely on other factors to ease this need, such

as various light conditions [15, 16] or multiwavelength [17].”

- Intro sentence starting “However, because of...” is not clear.

In the original expression, we wanted to state that such method needs to scan around to determine the depth and glass points. They mainly focus on 2D route mapping and is not suitable for getting the whole 3D image of the glass scene. We have modified this sentence in the third paragraph of Introduction (Page 1):

“However, these methods require to scan around to determine the depth and glass points, thus they are still in the stage of 2D route mapping.”

- Intro “surrounded by some borders” also not clear. There are quite a few sentences like this that will not make sense to most readers.

We mean that the areas with frames or borders (such as an empty frame without glass) are easy to be misjudged as the reflective surfaces. We have refined this sentence in the fourth paragraph of the Introduction (Page 1):

“But we find that even the recent works [36, 37] might still misjudge the areas with frames or borders, such as an empty frame without glass, as the reflective surfaces.”

And the analysis in Fig. 3e-3h can also illustrate this effect.

- Define “polarization selection ratio”.

We have added the power curve and OPSR curve of the HCG-VCSEL array in Supplementary Section I, where the definition of “polarization selection ratio” and characterization of the HCG-VCSEL array are well clarified.

- Another couple of sentences to explain the core principle of your method is needed in the intro.

Thanks for the advice. We have added the explanation of our method in the last paragraph of Introduction (Page 2):

“According to Fresnel theory [39], the specular reflection from a reflective surface maintains the same polarization as the incident polarized light. However, diffused reflection from objects without smooth surfaces do not exhibit any polarization even incident by a strongly polarized light. Hence, using a polarization-selection CMOS camera can differentiate reflection from a reflective surface. Thus, the depth information of a reflective surface or objects behind it can be obtained based on choice of the polarizer direction. Here, we report three experiments to demonstrate how PSL can be used to see as well as to see through highly reflective surfaces.”

- The captions are huge and repeat much text from the main body. They should be much shorter.

Thanks for the advice. We have shortened and refined all the captions in the revised manuscript.

- In section 2, it is clear about how the reflective surfaces are detected. But more information needed on how depth is calculated.

The depth of the reflective surfaces can be obtained by calculating the spatial displacement of the reflected dot patterns. New sections in Methods and Supplementary Information are added to describe this principle:

“Calculation of the depth of reflective surfaces.

By calculating the spatial displacements of known dot patterns, depth information of reflective surfaces can be obtained according to triangulation. The received dot patterns on the reflective surfaces are shown in Supplementary Fig. S20 and a model of triangulation is also shown to illustrate how depth is calculated.”

- The effect of natural noise depends on incidence angle. This is glossed over.

As mentioned above, we have added a new case, i.e., the scene of wall corner, to analyze the impact of reflective noise. In this case, comparisons in different heights (i.e., different incident angles) with stereo vision camera and ToF camera (Fig. 5d-5l and Supplementary Fig. S13) are added to show that our PSL 3D sensor can obtain the correct depth information with its P-polarized setting at all heights while the other two cameras get the wrong depth measurements due to the reflection and multipath noise. The corresponding main text content is the third paragraph of the section “Results of seeing against the reflective noise” (Page 6-8, “In the next case, ... with this P-polarized setting”)

- I’m confused about the geometry considered for fig 2. Maybe add a diagram?

The original Fig. 2 is now changed to Fig. 3. In this figure, we have added a diagram (Fig. 3a) to describe the geometry. In this scene, the sensor and the persons are on the same side, whose distances to the glass are 1.2m. The stereo vision camera provides erroneous depth measurements (Fig. 3c) based upon the reflected image of two persons, while our PSL 3D sensor can get the correct depth of the glass (Fig. 3d).

Figure 3: Analysis of 3D imaging in scenes with reflective surfaces. (a) and (b) are scheme and picture of the indoor glass wall scene. The glass wall is about 1.2m in front of the sensor, with a reflecting image of two persons and the sensor. Note that two persons are on the same side as the sensor. (c) Depth image from the stereo vision camera (Intel RealSense D455). (d) Depth image from the PSL 3D sensor.

Reviewer #2 (Remarks to the Author):

Polarization imaging has been studied and applied for a long time, always with a polarizer after the source to generate the polarized light. The paper entitled “Polarization Structured Light 3D Depth Image Sensor for Scenes with Reflection Surfaces” presents a 3D sensor with polarized structured light for reflection surfaces and shows several examples of seeing glass surfaces. The PSL camera includes an HCG-VCSEL array with an intrinsic high polarization selection ratio as the laser source without a discrete polarizer. The results in the paper are very interesting, and show promising applications. This work will stimulate the research of polarization imaging in 3D sensing. I have the following comments and questions.

We thank the reviewer for the neat summary and confirmation of our work. We have added the required measurements and revised our manuscript according to the suggestions. Please see the following responses.

1. In supplementary Fig. S4, it is better to give a schematic to show “measure the working range of the sensor by rotating it in the horizontal and vertical directions”. It is not easy to understand the measurement to the readers.

Thanks for the advice. The original Supplementary Fig. S4 is now changed to the Fig. S16 in the Supplementary Section IX. We have added the diagrams and corresponding results of measurements in Supplementary Fig. S16.

2. In the last row of Fig. 5, What does the red region present? “In the completed point cloud, the glass is displayed in red.”, I can’t understand why the authors add a red region in a tilted style in the point cloud.

The original Fig. 5 is now changed to Fig. 6. The red region in the last row is the completed glass. As you can see in the principle of completing the reflective surface (Fig. 2), in “Before”, only the middle region of glass is detected. After applying step 1 and step 2, the whole glass area can be completed in “After”. Thus, this new completed glass is highlighted in red. This is also the same for the results of Fig. 6 and Supplementary Fig. S14. However, in order to show the point cloud of both the glass and objects behind, they are displayed in different viewing directions in the rows of “Before” and “After”.

Thanks for raising this question. In order to show it more clearly, we have modified the “Before” and “After” in step 2 of Fig. 2 as well as those in Fig. 6 and Supplementary Fig. S14. In addition, we have put Fig. 2 to the subsection “Completing the reflective surface” in section “2. Principles”, showing the completing method together with other principles.

Step 1

Step 2

Figure 2: **Principle of completing the reflective surface.** In **step 1**, the glass region is extracted through a combination of depth and color channels, where the glass points are highlighted in red. In depth channel, two depth images are obtained in polarization 0° and 90° respectively. Then, subtraction is employed between these two depth images for extraction of the glass points. In color channel, the glass boundary is predicted from the color image of the scene using deep learning method. In **step 2**, the extracted glass points are transformed to the world coordinate for fitting. After fitting, the results are mapped to the color channel. In this channel, the reflective surface can be interpolated and completed inside the glass boundary. At last, the completed point cloud can be converted back to the world coordinate, as shown in the before and after comparison on the right side.

3. In this polarization imaging application, one challenge is the polarized laser source. Previously, a polarizer is used after the laser source to generate the polarized light. One novelty in this work is using the HCG-VCSEL array with a high polarization selection ratio as the laser source, eliminating the discrete polarizer. In the supplements, two papers are cited to show the spectral high polarization selection ratio of 30 dB. Still, I think the power curve and the power OPSR curve of the HCG-VCSEL array should be included in the supplement. Since the thermal crosstalk exists in the array, is the power OPSR of the HCG-VCSEL array the same as that of the single HCG-VCSEL? Can the authors analyze the influences of the polarization ratio on the image quality of the PSL

camera? The power of the VCSEL array in the 3D sensing can affect the depth accuracy. How about the output power of the HCG-VCSEL array used in the experiments?

Thanks for the confirmation of our novelty. According to the suggestions, we have measured the power curve and the OPSR curve of the HCG-VCSEL array, DBR-VCSEL array and the single HCG VCSEL. The results are added in **Supplementary Section I (Supplementary Fig. S1)**. All these VCSELs are from the same production patch. From the results, we can see that the VCSEL array possesses similar polarization characteristics as the single VCSEL. The description in the Supplementary is copied below:

“In Fig. S1a, we show the OPSR and power curve as the function of working current, where solid lines are the results of HCG-VCSEL array and dot lines are the results of DBR-VCSEL array. We can see that as the current increases, the OPSR of HCG-VCSEL reaches the plateau at around 17dB, meaning the output light of the HCG-VCSEL array possesses polarization. However, as for the DBR-VCSEL array, it keeps around 0dB, which has no polarization. From the power curves of two types of arrays, we can see that the top HCG does not affect the output power much, making the HCG-VCSEL array suitable for the TX source. In Fig. S1b, we show the OPSR curve and power curve of a single HCG VCSEL in the same fabrication patch. We can see that the OPSRs of the HCG-VCSEL array and the single HCG VCSEL are close and the power obeys the multiple relation, which means the array almost has no impact on the original characteristics. In this patch, because of the fabrication error, the OPSR is not as high as that of supplementary references [1, 2]. But this polarization ratio is sufficient to prove the superiority of our PSL 3D sensor in the application of 3D imaging, as demonstrated in the three experiments of the main text.”

Figure S1: OPSR and power curve of VCSELs. (a) OPSR and power curve of HCG and DBR VCSEL arrays. (b) OPSR and power curve of a single HCG VCSEL.

As for the influences of the polarization ratio on the image quality of the PSL camera, we use the DBR-VCSEL-based camera, which has no polarization, to illustrate the effect. The results are added in the last paragraph of **Supplementary Section V (Supplementary Fig. S8)**. We can see that the glass depth area in polarization 0° and polarization 90° is similar, and the function of elimination cannot be achieved in polarization 90° . Thus, for a camera without such high polarization ratio, it cannot

achieve the functions as our PSL 3D sensor.

Figure S8: **Results of DBR-VCSEL-based sensor.** (a) The picture of the scene. (b) Depth map from Pol 0°. (c) Depth map from Pol 90°. The yellow-black color bars apply to both (b) and (c).

In addition, as claimed in the last sentence of Supplementary Section I: “in order to control variables, we set the current of the PSL 3D sensor to be 1.5A at each experiment, which means the output power is around 2.2W.”

4. In Fig. 5, Column 2-4, Row 8, some point cloud regions are missing. What are the reasons? How to solve this issue?

Because our method is a single view of detecting and completing, it will have empty space if we show the point cloud in different viewing directions (e.g., top view, side view). If we want to reconstruct a complete 3D shape of an object, we need to employ our method in various angle.

Thus, we have added a new experiment in Supplementary Section X (Supplementary Fig. S17), in which we have achieved the 360° panoramic reconstruction of a fish tank scene. Specifically, we capture four sides of the fish tank using the PSL 3D sensor, as shown in the first row of Fig. S17a. By setting the polarization of the PSL 3D sensor to 0° and 90°, we obtain depth images at two different angles, as illustrated in the second and third rows of Fig. S17a. Consistent with previous experiments, when polarization is 0°, the PSL 3D sensor can reconstruct the glass part of the fish tank and the surrounding objects. When polarization is 90°, it can eliminate the glass part and retain only the surrounding diffuse components. Combined with the glass boundary in the fourth row, we then extract, fit and complete the glass, as shown in the fifth, sixth and seventh rows of Fig. S17a, respectively. Finally, by combining the reconstructions of the four sides of the fish tank with the ground as the reference plane, we use CloudCompare to fuse the multi-angle data and achieve the 360° panoramic reconstruction of the fish tank scene, as shown in the 3D color and grayscale schematics in Fig. S17b and S17c, respectively.

We have also mention this in Page 10 of the main text: “The above experiments of completion are from the single view. If we want to reconstruct the whole scene, we need to employ our method in different views. In Supplementary Fig. S17, we show the reconstruction of a fish tank by employing our method in four sides of the fish tank.”

Thanks for raising these questions, we have also modified the figures in the rows of “Before” and “After” in Fig. 6 and Supplementary Fig. S14 to show our results more clearly.

Figure S17: **Panoramic reconstruction of a fish tank.** (a) Detection and completion results of four sides of the fish tank. (b) and (c) are the panoramic reconstruction of the fish tank, rendered with color and in grayscale, respectively.

5. The PSL camera shows the advantage to see the reflection surface. I see in the

paper that, the error can reach 80 mm, and the average error of the entire plane is at the level of 10 mm. As far as I know, the current structured light 3D camera can reach an accuracy of 1 mm at 1-2 m distance. The results seems that the PSL camera has no competition in the depth accuracy, compared with the current structured light camera for 3D sensing. How to achieve an accuracy of 1 mm at 1-m distance with the PSL camera?

Thanks for raising this issue. The reasons why we get the high average error and the corresponding solving methods are:

- 1) In our previous experiments, we got the benchmark by covering the entire plane with a wide piece of paper or cloth, which would be loose on the surface. This made the average error of the entire plane high. Thus, in our new experiments, we use five thin stickers to place on the reflective surfaces, as shown in the following Fig. R1a. We get the measurement of these five stickers and fit the plane based on these depths. Then we use this fitted plane as the benchmark to calculate our measurement accuracy.
- 2) In our previous measurement setup, as you can see in Fig. R1b, the RGB camera is blocked by the polarizer holder. So, before measuring the benchmark, we needed to get the polarizer holder out by loosening its screws carefully in order to obtain the RGB image of the scene. This operation would make the sensor move slightly. Hence the previous benchmark did not align well with the completing results, making the error large in the edge. For solving this issue, we use 3D printing to make a new polarizer holder, which is hollow at the place of the RGB camera (as shown in Fig. R1c). Every time we take picture of the scene, we only need to take out the polarizer easily, which sticks to the holder by magnets.

With these solutions, we have redone all the experiments in Fig. 6 and Supplementary Fig. S14 and remeasured the absolute error between the completed reflective surfaces and the benchmarks. The new error analysis is shown in Fig. 7. From the results, we can see that the errors maintain at the level of 1mm, demonstrating the precise reconstruction of our method. There are outliers in the error maps. These errors mainly come from the fluctuation of the data, like the influence of the interreflection, which can be reduced to a much lower level if we include the model to suppress it in the future.

Figure R1: **Schemes for explaining the solutions to the high errors.** (a) The depth map of the benchmark. (b) The original measurement setup. (c) The new measurement setup.

Figure 7: **Error analysis.** (a)-(h) are the error maps of eight scenes of reflective surfaces, which are

the absolute errors between the completed reflective surface and the fitted benchmark. (i) The boxplot graph of these eight error maps. It includes the box, mean value and outliers of each map, showing their statistical properties.

The corresponding content in the main text is “As for the measurement accuracy, ... in the future.” (Page 8-10, seventh paragraph of the section “Results of completing the reflective surface”).

6. In the conclusion section, I suggest to give some discussions on the potential application fields of the PSL camera.

Thanks for the suggestion. We have added the discussion on the potential application fields of our PSL 3D sensor in the last paragraph of section “Discussion” (Page 11): “Serving as the eyes of a robot, PSL 3D sensor will find many applications such as service robots and logistics robots. These robots will inevitably encounter the problem of reflective surfaces such as the glass door and floor corner shown in the main text. The PSL 3D sensor can make them handle these cases correctly. Besides, object or scene reconstruction of reflective surfaces, like office and exhibition hall, will also benefit from this method. Thus, with the special polarization characteristic, the PSL 3D sensor can be further extended to a wide range of indoor and outdoor applications.”

7. This paper presents the results of PSL camera with HCG-VCSEL array. Another approach of 3D sensing is time of flight (TOF) method. I suggest that the authors give some discussions on the TOF polarized imaging with HCG-VCSEL array.

Thanks for the suggestion. According to the suggestion, we have added the comparison with the ToF sensor in two experiments:

First, in the experiment of seeing against the reflective noise, we compare our PSL 3D sensor with the ToF sensor in the case of an indoor wall corner scene, whose experimental setups are displayed in Fig. 5 of the main text. In this case, comparisons are carried out in different heights (i.e., different incident angles) and the results of the ToF sensor are shown in the following Supplementary Fig. S13. From the results, we can see that the ToF sensor is unable to obtain the correct depth information at all heights. In detail, the floor part in the point cloud (pointed by red dot box and solid arrow) penetrates into the plane because of the impact of noise while the PSL 3D sensor works well at all heights with its polarization settings.

Figure S13: **Detecting results from ToF sensor.** (a)-(c) are depth maps at different heights. (d)-(f) are their corresponding point clouds. The jet colorbar applies to (a)-(c).

Second, in the experiment of completing the reflective surface, we have analyzed the performance of the ToF sensor and our PSL 3D sensor. The comparison results are added in Supplementary Section XII (Supplementary Fig. S19). The revised content related to this is in the last paragraph of the section “Results of completing the reflective surface”: “The ToF sensor is also compared in Supplementary Fig. S19, which once again demonstrates the distinction of the PSL 3D sensor.”

And the description of this experiment in the Supplementary is copied below:

“In this comparison, we place the sensor 51cm directly in front of the glass (Fig. S19a). Since the purchased ToF sensor (Orbbec Femto W) has been integrated together, we cannot replace the source with an HCG VCSEL, so we add polarizers to its TX and RX for testing. As shown in Fig. S19b, TX is set to pol 0° and RX is set to two directions. We also test the PSL 3D sensor in the same scene. The RGB pictures of the scene obtained by the ToF sensor and PSL 3D sensor are displayed in Fig. S19c and Fig. S19d respectively, where the diffuse sticker on the glass is denoted by orange dot line.

In this experiment, the results of ToF sensor are shown in Fig. S19e to S19h. As seen from Fig. S19e, the area of glass that can be detected is very small (denoted by white dot line), even though neither TX nor RX has a polarizer added. In addition, the depth of the sticker on the glass is denoted by orange dot line. When we add the polarizer of pol 0° to TX, the glass and sticker can be measured (Fig. S19f). When we add a polarizer of 0° to the RX, both the glass and sticker can still be detected because of the same polarization of TX and RX (Fig. S19g). Although the tree and wall behind are absent because of the occlusion, these are irrelevant factors, and we only need to focus on the parts of glass and sticker. When we place the polarizer of RX in pol 90° , the glass part is eliminated in the depth map because its reflection maintains the same polarization, while the diffuse sticker is detected in the depth map (Fig. S19h).

As for the PSL 3D sensor, as shown in Fig. S19i, we get a wide range of glass depth when we set it to polarization 0° . If we change to polarization 90° (Fig. S19j), the glass part can be removed, while the sticker on the glass is retained because of the diffuse

reflection.

Although the ToF sensor can also add polarization characteristics to detect and eliminate the glass depth, it cannot obtain enough glass depth information because the depth calculation method is different from that of structured-light-based method. Therefore, PSL 3D sensor remains a powerful tool for the task of detecting and completing reflective surfaces.”

Figure S19: **Comparison with ToF sensor.** (a) Side view of the scene. (b) Setup of the ToF sensor. (c) Color image of the scene from the ToF sensor. (d) Color image of the scene from the PSL 3D sensor. (e)-(h) are depth maps from the ToF sensor, where their experimental settings are shown on the top of each figure. (i)-(j) are the results from the PSL 3D sensor.

Reviewer #3 (Remarks to the Author):

1. It can be seen from the method part of the paper that the author's understanding and use of polarization optics is relatively simple, so it is suggested to learn relevant knowledge;

We think the structure of our previous manuscript is not organized so well that the reviewer cannot understand our approach. Thus, we have polished all the method parts and arranged them to the second section (Fig. 1 and Fig. 2 in the revised manuscript), including the composition of our PSL 3D sensor, the working principles of the three experiments: seeing through the reflective surface, seeing against the reflective noise, completing the reflective surface.

In the experiment parts, a lot of additions and modifications have been made. They include the systematic analyses in the influence factors for seeing through the reflective surface (Table 1, Fig. 4 of the main text, Supplementary Fig. S5-S8 and Movie 1), the new comparisons with stereo camera and ToF sensor in different scenes and different heights (incident angles) (Fig. 5 of the main text and Supplementary Fig. S9-S13), and the high precision achieved in detecting and completing the reflective surface (Fig. 6-7 of the main text and Supplementary Fig. S14, S17-S19). All these new results and strong evidences have fully demonstrated the feasibility and wide application of our method.

2. As a device with similar functions to kinect, there have been relevant studies similar to this paper since 2009. Please carefully investigate relevant literatures;

Thanks for the advice. After careful investigation, we have found that a relevant study that employs the fusion of Kinect and sonar to obtain the depth images of glass scenes [Zhang, Y. et al. IEEE Transactions on Pattern Analysis and Machine Intelligence **40** (8), 1785-1798 (2018)]. We have cited this work (Reference [33]) in the Introduction. The added content is in the third paragraph of the Introduction (Page 1): “As for structured light projector, ... make the system costly, huge and complex.”

However, due to the sparse data and narrow angle of the sonar, it needs to sweep multiple times to obtain enough information. Besides, the fusion with sonar will make the system costly, huge and complex.

In contrast, our PSL 3D sensor is a single view method. With the special polarization characteristic, we can obtain enough depth information of the reflective surface in Polarization 0° with single shot and remove it in Polarization 90°. In order to prove the feasibility, we have prepared extensive experiments, as shown in Table 1, to systematically analyze the performance of our PSL 3D sensor. From the new results (Fig. 4 of the main text and Supplementary Fig. S5-S8, the corresponding contents are the whole section of “Results of seeing through the reflective surface ...” (Page 5-6)), it is proved that regardless of different incidence angles, or different distances and object densities, the PSL 3D sensor can eliminate the depth of the reflective surface and achieve the function of seeing through the reflective surface. This distinction can be further used to the experiment of completing the reflective surface.

Table 1: Influence factors for seeing through the reflective surfaces

Influence factor	Specific experimental condition		
Distance from 3D sensor to glass	0.4m	0.8m	1.2m
3D sensor incident angle	-30°	0°	30°
Distance of objects behind glass	0.3m	0.5m	0.7m
Density of objects behind glass	low	medium	high

In addition, our PSL 3D sensor can also be applied directly to detect against noise, as demonstrated in the new comparisons in Fig. 5 of the main text and Supplementary Fig. S9-S13 (the corresponding contents are the whole section of “Results of seeing against the reflective noise ...” (Page 6-8)). All these are not possible for the traditional methods, showing the speciality of our PSL 3D sensor.

3. In the past 20 years, there has been a lot of work using polarization optics to separate specular reflected light and carry out depth estimation. The algorithm and hardware equipment in this paper have no innovation, and the separation effect is not good, which has no advantage compared with relevant achievements after 2010;

Thanks for raising this issue. After careful investigation, we have cited the relevant works and give some discussions in the second paragraph of the Introduction (Page 1). The modification is copied here:

“However, due to the inability to differentiate mirages from real objects, stereo vision will result in a virtual and wrong distance. Some methods use multiple polarized images [18-22] or local reflection cues [23, 24] to separate the reflection. In some cases, an additional infrared sensor is used to provide depth information and to remove the mirage images [25, 26]. One major drawback of polarization-stereo-vision methods is that they do not result in correct depth measurements of the reflective surfaces (e.g., the depth of the glass).”

From your remarks, we have realized that previous experiments in seeing against reflective noise are inadequate. Therefore, in our revised manuscript, we have carried out comparisons with stereo camera and ToF camera in different scenes and different heights (incident angles) to prove the advantages of our PSL 3D sensor. All these new results (Fig. 3a-3d and Fig. 5 of the main text, Supplementary Fig. S9-S13) can demonstrate that our PSL 3D sensor has better performance than other cameras and can be directly applied to these scenes.

4. This paper uses the form of rotating polarizer to acquire polarized light, which is a time sequence acquisition method, its application scenario is greatly limited, and its practical application value is low;

Thank you for raising this issue. As for the improvement of the polarizer, in the future work, “we can design a unique polarizer array in the receiving end like that in [Millerd, J. et al. Fringe 2005, 640–647 (Springer, 2006)] (Reference [44]), which can be coated on the CMOS camera and receive the signal from different polarizations simultaneously,

making our system more compact.” We have added this discussion in the third paragraph of section “4. Discussion”.

5. There is no literature related to polarized light separation specular reflection component in the references, which reflects that the author has no knowledge accumulation in this field and does not understand the latest relevant research results;

According to the remark, we have added the related works and given some discussions in the Introduction, as in the reply to point 3 above.

6. The paper lacks the introduction of mathematical model and corresponding schematic diagram (but the model of pol0-pol90 is too simple, so the reviewer does not expect the summary diagram of this model);

Although the reflected light is affected by the randomly distributed particles on the reflective surface, expanding the reflection range to a certain extent, this kind of reflection is not diffuse reflection and will still maintain the original polarized direction if we project polarized light on it. The model is also described in [Torrance, K. E. & Sparrow, E. M. Theory for off-specular reflection from roughened surfaces. *J. Opt. Soc. Am.* **57**, 1105-1114 (1967).] and [Wolff, L. B., Nayar, S. K. & Oren, M. Improved diffuse reflection models for computer vision. *International Journal of Computer Vision* **30**, 55–71 (1998).], which are also cited in our revised manuscript (Reference [42, 43]). Thus, when TX and RX have orthogonal polarized directions, the information belonging to the reflective surface will be completely eliminated.

In order to prove it, as mentioned in the responses to point 1, 2 and 4, we have added a large number of experiments under different conditions (Table 1). The new results (Fig. 4 of the main text and Supplementary Fig. S5-S8) shows that our PSL 3D sensor can achieve the function of detecting and eliminating the depth information belonging to the reflective surface, also called as seeing through the reflective surface, which is not possible with previous methods. We have also provided a Supplementary Movie to support this, where the depth of the reflective surface realizes from appearance to disappearance by rotating the RX polarizer. In addition, in the experiment of completing the reflective surface, such distinction is utilized to achieve reconstruction of the reflective surfaces with high precision in different scenes (Fig. 6-7 of the main text and Supplementary Fig. S14, S17-S19). All these new results attest to the correctness of our model.

7. In the noise experiment to remove the reflection of natural light, what should be done if the reflected noise area overlaps or partially overlaps with the measured target? In the experiment of this paper, the two are separated from each other, which is a special and simple case. The author did not solve the key problems mentioned above.

Thanks for raising this issue. From your remarks, we have realized that previous experiments in seeing against reflective noise are inadequate. Therefore, in our revised manuscript, we have carried out comparisons with stereo camera and ToF camera in

different scenes and different heights (incident angles) to prove the advantages of our PSL 3D sensor.

In the case of glass scene, three kinds of outdoor situations (Fig. 3a-3d, Fig. 5a-5c and Supplementary Fig. S9a-S9c) are used to show that the reflective noise, like sunlight and shadow of objects, incapacitates the stereo vision camera for obtaining the depth of the scene, while the PSL 3D sensor can measure them clearly. To be detailed: (1) In the case of normal incidence (Fig. 3a-3d), which is also shown in our previous manuscript, the reflection noise of two persons makes the stereo vision camera provide erroneous measurement while our PSL 3D sensor obtain correct depth of the glass. (2) In the case of sunlight noise (Fig. 5a-5c), the sunlight noise leads to black-irregular-patch errors in the stereo vision camera while our PSL 3D sensor can measure the scene clearly. (3) We have added another case of outdoor scene (Supplementary Fig. S9a-S9c), which we think is the situation mentioned in the remarks. In this situation, the reflective noise on the glass overlaps with the position of the person to be detected, which causes the stereo vision camera to fail to obtain the correct depth of the person. But for the PSL 3D sensor, the correct depth information can be obtained through the polarization combination of TX and RX.

Figure 3: **Analysis of 3D imaging in scenes with reflective surfaces.** (a) and (b) are scheme and picture of the indoor glass wall scene. The glass wall is about 1.2m in front of the sensor, with a reflecting image of two persons and the sensor. Note that two persons are on the same side as the sensor. (c) Depth image from the stereo vision camera (Intel RealSense D455). (d) Depth image from the PSL 3D sensor.

Figure 5: **Seeing against the reflective noise.** (a)-(c) are comparison results of an outdoor glass scene, where (a) is the picture of the scene, (b) is the depth map from stereo vision camera (Intel RealSense D455) and (c) is the depth map from PSL 3D sensor.

Figure S9: Comparison on an outdoor glass scene. (a) Scene of the experiment. (b) Depth map from the stereo vision camera. (c) Depth map from the PSL 3D sensor.

In the case of wall corner scene, comparisons in different heights (i.e., different incident angles) with stereo vision camera and ToF camera (Fig. 5d-5l and Supplementary Fig. S13) are added to show that our PSL 3D sensor can obtain the correct depth information with its P-polarized setting at all heights while the other two cameras get the wrong depth measurements due to the reflection and multipath noise.

Moreover, experiments of polarization combinations are added in Supplementary Fig. S10-S12 to show the advantages of the PSL 3D sensor. Through all these common scenes of reflective surfaces, our PSL 3D sensor is proved that it can filter noise by specific polarization combinations, improve the signal-to-noise ratio and obtain complete depth information of the scene.

Figure 5: **Seeing against the reflective noise.** (d)-(l) are comparison results of a indoor corner scene. The experiments are done in different height, whose schemes are shown in (d), (g) and (j). The corresponding depth map and point cloud from stereo vision camera are displayed in (e), (h) and (k) respectively, while those from PSL 3D sensor are in (f), (i) and (l). The jet colorbar with range (0.48m, 1.1m) applies to (b), (c), and the one with range (0.3m, 1m) applies to the depth map column.

Figure S13: **Detecting results from ToF sensor.** (a)-(c) are depth maps at different heights. (d)-(f) are their corresponding point clouds. The jet colorbar applies to (a)-(c).

8. Removing specular reflection by rotating the polarizer is not accurate. Specular reflection is local information, which is related to incident light Angle, etc. The difference between two images cannot guarantee the detection and removal of reflection area. If there is a strongly reflective object behind the glass, the device will fail.

Thanks for raising this issue. As replied in point 6, we have illustrated the model and added extensive experiments to prove that our PSL 3D sensor can achieve the function of seeing through the reflective surface.

As for the complicated situation of **a strongly reflective object behind the glass**, we have added a new experiment in **Supplementary Section XI (Supplementary Fig. S18)** to show that our method can work robustly in this complicated situation.

“In this section, we experimentally verify the reconstruction ability of the PSL 3D sensor when there is another strongly reflective object (such as a mirror) behind the glass. In this scene, the PSL 3D sensor is placed 1m in front of the glass door, and a rectangular mirror and a cabinet are placed 0.5m behind the glass door, as shown in Fig. S18a and S18b. We use the PSL 3D sensor to obtain two depth images at polarization 0° and 90° , as shown in Fig. S18c and Fig. S18d. At polarization 0° , we are able to obtain the depth of the front glass (red dot line) and the back mirror (white dot line). At polarization 90° , these two parts are eliminated simultaneously and only the surrounding diffuse components are remaining. After applying subtraction, the depths of the glass and mirror are both retained, as shown in Fig. S18f. But our main purpose is to identify the front obstacle. Hence, we can filter out the reflection from the back, making it successful to determine the front glass area, which is demonstrated in Fig. S18g. The final completed result is shown in Fig. S18i. As seen from the error map in Fig. S18j, our method can achieve reconstruction with high precision of 2mm on the average”

Figure S18: **Detection and completion result in scene with glass and mirror.** (a) Scheme of the experiment. (b) RGB image of the scene. (c) Depth map at polarization 0°. (d) Depth map at polarization 90°. (e) The predicted boundary. (f) The subtraction count. (g) The extracted reflective surface region. (h) Fitting of the extracted points in the world coordinate. (i) The completed depth image. (j) The calculated error map.

Reviewer #2 (Remarks to the Author):

The authors have carefully addressed my comments and questions, and made new additions. Therefore, I recommend that the manuscript should be accepted for publication in Nature Communications.

One more minor suggestion: The authors add the information about the driving conditions of the VCSEL and VCSEL array in Figure S1: continuous-wave current or pulsed current (period and duty cycle), in the final manuscript.

Reviewer #3 (Remarks to the Author):

The authors of this paper have designed a three-dimensional sensor based on polarization structured light, aiming to achieve depth perception through highly reflective surfaces. But there are still some issues should be solved:

1. Insufficient proof of utility: There are many polarization imaging-based transparent target recognition methods, anti-reflection methods, and 3D reconstruction methods. What is the difference/innovation between the method in this paper and them? If the existing method is used to remove reflections and then perform depth estimation, what defects will there be compared with the method in this paper? Experiments are too lacking in comparison. The experiments in the main text and supplementary material are still not convincing.

2. Insufficient proof of utility: Removing specular reflection by rotating the polarizer is not accurate. The author considers a curved surface. However, the reflection removal of flat media is completely different from that of curved surfaces. How the author did it is really doubtful. The author's experiment is considered too simple and does not have enough algorithm and theoretical support.

3. Deeper issues not addressed: It is not mentioned whether the authors have delved into deeper issues related to highly reflective surfaces, such as sensor performance in complex environments, possible limitations, and how it could be combined with other techniques to improve effects etc.

Reviewer #4 (Remarks to the Author):

This Ms has been seen and judged already by 3 reviewers. An enormous number of improvements have been suggested. The authors replied to all of them with care and in much details, such that the 24 pages file with comments and replies is now longer than the Ms.

All 3 reviewers judge, that the subject is of highest relevance and might be published after revisions.

After reading all the material I judge, that the Ms is now ready for publication and does not need any more revisions

Response to Reviewers' Report: NCOMMS-23-10384A

We appreciate the editors' and reviewers' time and efforts in reviewing our manuscript entitled "Polarization Structured Light 3D Depth Image Sensor for Scenes with Reflective Surfaces". According to the suggestion of reviewer #2, we have added the description of the driving condition. As for the comments of reviewer #3, we have made the clarifications in the point-by-point response.

In the following, we outline the *reviewers' comments (in italic)* and the responses to those specific comments (in regular text). Hereinafter we referred to the revised manuscript and the previous response letter as **the Manuscript** and **the Response**, respectively.

Reviewer #2 (Remarks to the Author):

The authors have carefully addressed my comments and questions, and made new additions. Therefore, I recommend that the manuscript should be accepted for publication in Nature Communications.

One more minor suggestion: The authors add the information about the driving conditions of the VCSEL and VCSEL array in Figure S1: continuous-wave current or pulsed current (period and duty cycle), in the final manuscript.

We appreciate the recommendation for publication.

According to the suggestion, we have added the driving condition in the last part of Supplementary Section I: "the driving condition is pulsed current with 33 microseconds in period and 0.03 to 0.13 in duty cycle, namely 1 microsecond to 4.3 microseconds in pulse width."

Reviewer #3 (Remarks to the Author):

The authors of this paper have designed a three-dimensional sensor based on polarization structured light, aiming to achieve depth perception through highly reflective surfaces. But there are still some issues should be solved:

"Achieving depth perception through highly reflective surfaces" is BUT ONLY ONE of the THREE merits of our PSL 3D sensor.

Our PSL 3D sensor is able to perform THREE functions: (1) see through the reflective surface, (2) see against the reflective noise and (3) complete the reflective surface. This is clearly elaborated in **the Response**. Our replies to points raised by reviewer #1 and that to point 1 of reviewer #3's comments can be found on pages 1-4 and page 17, in **the Response**, respectively. In addition, we have revised **the Manuscript** substantially with many additions and modifications in the principles and experiments to attest to this fact.

1. Insufficient proof of utility: There are many polarization imaging-based transparent target recognition methods, anti-reflection methods, and 3D reconstruction methods. What is the difference/innovation between the method in this paper and them? If the existing method is used to remove reflections and then

perform depth estimation, what defects will there be compared with the method in this paper? Experiments are too lacking in comparison. The experiments in the main text and supplementary material are still not convincing.

Regarding the concern on the novelty of our work, we have already added citations as well as comparisons with other methods, as explained in **the Response**. We modified the “Introduction” section of **the Manuscript** to include discussions of shortcomings of conventional techniques and benefits of the novel PSL 3D sensor. They can be found in the second to fourth paragraphs on page 1 of **the Manuscript**, as well as in **the Response** (point 2 in pages 17-18, point 3 in page 18 and point 7 in pages 19-22 in the previous report of reviewer #3).

2. Insufficient proof of utility: Removing specular reflection by rotating the polarizer is not accurate. The author considers a curved surface. However, the reflection removal of flat media is completely different from that of curved surfaces. How the author did it is really doubtful. The author's experiment is considered too simple and does not have enough algorithm and theoretical support.

We have described the theoretical model and replied in **the Response**, addressing points 1, 2 (pages 17-18) and point 6 (page 19) of reviewer #3's first review.

We added a large number of experiments under different conditions to attest to the correctness of our model (Table 1, Fig. 4 of **the Manuscript and Supplementary Fig. S5-S8**, Fig. 6-7 of **the Manuscript and Supplementary Fig. S14, S17-S19**).

A Supplementary Movie is also added to demonstrate that the reflection from the glass panel can be eliminated by varying the polarizer angle.

Contrary to the statement of “*The author considers a curved surface*”, all the results have clearly shown that most of the experimental situations are wide and flat reflective surfaces, except the one in the third column of Fig. 6. This one demonstration in the ball-shaped glass is used to show that our method can also be applied to the situation of curved surfaces, proving the general applicability of our method.

3. Deeper issues not addressed: It is not mentioned whether the authors have delved into deeper issues related to highly reflective surfaces, such as sensor performance in complex environments, possible limitations, and how it could be combined with other techniques to improve effects etc.

We have addressed this question with substantial new data added to **the Manuscript and the Response**.

To address the complex environment issue, we have added substantial experiments to cover different experimental environments. We believe the 16 detection conditions in experiment 1, the 2 different scenes with multiple incidence angles in experiment 2, the 8 different reflective surfaces in experiment 3 and those in Supplementary Fig. S17-S18, are sufficient to represent various environments of reflective surfaces in our daily life.

These results also can be found in **the Response**. Our replies to point 2 and minor point 9 of reviewer #1's comments are on pages 1 and 6 of **the Response**, respectively. The replies to points 2, 4 and 5 of reviewer #2's comments are found on pages 7, 10 and 11 of **the Response**, respectively. And the reply to point 8 of reviewer #3's first review is

on page 23 of **the Response**.

As for the improvements, we have already given the discussion on the improvements and potential application fields of our PSL 3D sensor according to point 6 of reviewer #2 and point 4 of reviewer #3, on pages 14 and 18 of **the Response**, respectively.

Reviewer #4 (Remarks to the Author):

This Ms has been seen and judged already by 3 reviewers. An enormous number of improvements have been suggested. The authors replied to all of them with care and in much details, such that the 24 pages file with comments and replies is now longer than the Ms.

All 3 reviewers judge, that the subject is of highest relevance and might be published after revisions.

After reading all the material I judge, that the Ms is now ready for publication and does not need any more revisions.

Thanks for the recognition of our work and we appreciate the recommendation for publication.